



1    **Marine climate change over the eastern Agulhas Bank of South Africa**

4                                     Mark R Jury

5           Geography Dept, University of Zululand, KwaDlangezwa 3886, South Africa

6                and Physics Dept., University of Puerto Rico Mayaguez, USA, 00681

7                              submitted to EGU ocean Sci 4 June 2020

**Abstract**

12          The rate of change in the marine environment over the eastern Agulhas Bank, along the

south coast of South Africa (32-37S, 20-30E) is studied using reanalysis observations 1900-2015
and coupled ensemble model projections 1980-2100. Outcomes are influenced by resolution and
time-span: ~1 degree datasets covering the whole period capture large-scale changes, while ~0.5
degree datasets in the satellite era better distinguish the cross-shelf gradients. Although sea surface
temperatures off-shore are warming rapidly (.05°C/yr since 1980), a trend toward easterly winds
and a stronger Agulhas Current have intensified near-shore upwelling (-.03°C/yr). The sub-tropical
ridge during summer is drawn poleward by global warming and high phase southern oscillation
index. Cooler inshore sea temperatures suppress latent heat flux and contribute to coastal desicca-
tion (-.005 mm day$^{-1}$/yr) and vegetation warming (.1°C/yr) since 1980. Coupled ensemble projec-
tions from the Hadley and European models indicate that the shift toward drier weather and easterly
winds may be sustained through the 21$^{st}$ century.
Key words: South African, coastal climate change
mark.jury@upr.edu



# Introduction

The eastern Agulhas Bank along the south coast of South Africa is characterized by sharp gradients between inshore upwelling and an offshore current that advects warm water polewards at ~1 m/s (Lutjeharms et al. 2000). Downstream widening of the shelf and cyclonic shear causes uplift at the shelf edge (Schumann et al. 1982, Lutjeharms 2006, Goschen et al. 2015, Malan et al. 2018). Westerly and easterly wind regimes during winter to summer respectively induce alternating spells of downwelling and upwelling (Schumann and Martin 1991, Schumann 1999). Numerous small rivers discharge into the shelf zone (Schumann and Pearce 1997, Scharler and Baird 2005, vanBladeren et al. 2007). The inshore environment and large embayments (Fig 1a) are characterized by weak circulations and seasonal warming, and become stratified and productive during austral summer (Roberts 2010, Pattrick et al. 2013). The Agulhas Current meanders a few times per year (Goschen and Schumann 1990, Rouault and Penven 2011), while the mid-latitude jet stream meanders a few times per month advecting coastal lows and continental shelf waves along the shelf (Jury et al. 1990, Schumann and Brink 1990). Amidst these rapid changes are rising sea levels (Mather et al. 2009; Jury 2018) and longer summers.

The marine climate of southern South Africa is shaped by its plateau and sub-tropical latitude, producing a tendency for evaporation to exceed rainfall except during high-phase Pacific El Niño Southern Oscillation (ENSO), when regional atmosphere patterns couple with sea surface temperatures (SST) to enhance cyclonic storms (Philippon et al. 2012). Understanding trends in climate can inform resource management decisions and aid the socio-economic uptake of adaptive mitigating actions. Past research has found trends of .02°C/yr in air temperature (Kruger and Shongwe 2004, Morishima and Akasaka 2010, Jury 2013), however trends in other variables tend to be over-shadowed by short-term events and the sparsity of data before 1950 (Tadross et al. 2005, MacKellar et al. 2014, Kruger and Nxumalo 2017).

The main objective of this research is to establish the rate and pattern of observed and pro-

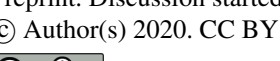


jected marine climate trends along the south coast of South Africa from 1900 to 2100. Scientific
questions include: 1) How has the wind field responded to a poleward shift of the subtropical ridge,
and how does that affect the shelf temperature and currents? 2) If the coastal ocean is cooling due to
wind- and current-driven upwelling, in contrast with the offshore environment – what are the con-
sequences for the local heat and water budget? 3) How does record length and dataset resolution
affect the result? and what is the impact of climate variability (eg. El Niño Southern Oscillation,
ENSO) on trend attribution? While the spatial focus is on the south coast of South Africa using
monthly datasets finer than 0.5°, context is provided at the large-scale using coarser model prod-
ucts.
**Methodology**

Modern data assimilation systems blend in-situ and ancillary measurements by iterating be-

tween climatology, persistence and theory, interpolating across gaps in time and space, and limiting
the influence of outliers. By reducing uncertainties, scientists now have a reliable means to evaluate
trends in marine climate. The monthly reanalysis products employed here include: ECMWF-int
atmosphere (Dee et al. 2011), ECMWF-20c atmosphere (Poli et al. 2016), ECMWF-ora4 ocean
(Balmaseda et al. 2013), NASA MERRA-2 atmosphere (Gelaro et al. 2017), NCEP CFSr-2 (Saha et
al. 2010), SODA-3 ocean (Carton et al. 2018), NOAA sea surface temperature (SST; Reynolds et
al. 2007), NOAA net outgoing longwave radiation (OLR; Lee et al. 2007), and NESDIS vegetation
temperature (Tucker et al., 2005). Table 1 lists the acronyms, data source, horizontal resolution and
time-span. Ocean-atmosphere fields with resolution finer than 50 km are capable of representing
cross-shelf gradients, and these are available in the satellite era 1980-2015. SODA-3 provides sub-
surface ocean data on temperature, salinity, currents and vertical motion; driven by MERRA-2
winds, multi-satellite altimeter and thermal measurements, blended with in-situ observations over
the shelf. Coupled land-atmosphere-ocean evolution is described by ECMWF products underpinned
by data assimilation (Hamrud et al 2015). Exploratory inter-comparisons of high-resolution NOAA
SST and surface temperature in the satellite-era reanalyses (CFSr-2, ECMWF-int, MERRA-2) show
coherent values and cross-shelf gradients, indicating they capture the inshore upwelling.

In addition to the monthly datasets, daily ECMWF-int sea level air pressure (SLP) fields

were analyzed using transient empirical orthogonal functions (EOF). The leading mode was deter-
mined and its spatial loading pattern and time score were analyzed for trends and cycling over the
period 1900-2015. Ship data, from the repository for marine data collected in South African waters:
SADCO, were analyzed in 0.1° bins for SST and wind speed, averaged 24.5-26.5°E 1950-2015 (cf.
Fig A1) and compared with 0.3° reanalysis products. Monthly river discharge records were ob-
tained for the Gamtoos and Sundays Rivers from the SA. Department of Water Affairs hydrology
service: SADW, and combined to understand the coastal hydrology.

The central method for analyzing marine climate change is to statistically regress a trend

line over a long record of temporal data, using the Pearson Product Moment least squares tech-
nique. The resultant slope and $r^2$ fit of the regression line provides a statistical way of determining
the rate of change or trend (signal) within the inter-annual fluctuations (noise). The temporal data
are filtered to annual and area-averages, according to the insights required.

Linear trends are spatially analyzed per grid point in three domains: large-scale map 45°-

20°S, 10°-50°E, regional-scale map: 32-37°S, 20-30°E, and depth sections over the shelf: 37-33°S,
averaged 24.5-26.5°E. Local trends are calculated by regression onto time series averaged over the
index area (35.5-33.5°S, 24.5-26.5°E) after reduction to annual and seasonal (Dec-Feb) interval. For
example, a sea temperature warming of 3°C over 100 years yields a .03 C/yr slope which is mapped
in relation to adjacent regression-fitted slopes. If year-to-year fluctuations reduce the $r^2$ fit below a
certain statistical threshold, then it is inferred that the signal is swamped by noise. Trends for U, V,
W wind and current components are calculated separately and combined into 'trend' vectors that
represent the slope or rate of change, as maps and sections. The $r^2$ fit of the trend is evaluated for
significance at 95% confidence. For long-term records having > 100 degrees of freedom, a mean-
ingful outcome requires $r^2$ > 4% (r > |0.2|). For satellite era records having < 40 degrees of freedom,




thresholds are reached at $r^2 > 9\%$ ($r > |0.3|$). Trends are embedded in noisy marine environments,
and so depend on time-span, local climate variability (Schlegel and Smit 2016), and quality of the
input data (Chaudhuri et al. 2013). The CFSr-2, ECMWF-int, and MERRA-2 reanalyses exhibit
similar trends (Kennedy et al. 2011, Decker et al. 2012) and yield comparable turbulent fluxes
around South Africa.
The trend of SST and zonal winds are analyzed by correlating the slope against its time se-
ries, for each month. For zonal winds, the index area is used (35.5-33.5°S, 24.5-26.5°E), but for
SST the analysis distinguishes between coast (33.8-33.9°S) and shelf-edge (34.9-35.0°S) latitude
bands. The resultant correlation values per month are plotted over the annual cycle to detect the
seasonality of trends in the period 1980-2015.
Using 18-month filtered values, hovmoller plots were constructed across the shelf to identify
how intra-decadal fluctuations mingle with climate change signals. After exploratory statistical
tests, a modulating influence was attributed to the Pacific southern oscillation index (SOI) or east-
west difference in SLP. Its time score is analyzed for trend and correlated with local SST and zonal
winds in annual and seasonal intervals 1980-2015. Similarly, a dipole mode is extracted by EOF
analysis of filtered ECMWF-esm projected SLP fields in the tropical Pacific, and temporal charac-
teristics are studied.
Projections of air temperature, precipitation and zonal winds from the coupled ensemble
ECMWF-esm v2.3 model (Taylor et al. 2012, Doblas-Reyes et al 2018) are analyzed over 1980-
2100 as large-scale trend maps and index-area time series. The simulation is forced by the rcp8.5
greenhouse scenario (vanVuuren et al. 2011, $CO_2$ +5 ppm /yr), and incorporates data assimilation in
the first 35 years that overlap with observations. Like most long-term projections, intra-member
dispersion is constrained by ensemble averaging and trends therefore emerge. The coupled ensem-
ble Hadley-esm model (Collins et al. 2011) is analyzed for zonal currents. Prior research found that
this model is one of the few to realistically represent ocean 'dynamic topography' (Jury 2018) and



sea level pressure fields around southern Africa (Dieppois et al. 2015). Its mean annual cycle of
zonal currents closely follows the reference Aviso-Copernicus product (cf. Fig A2).
An intercomparison of the reanalysis fields and model simulations is covered in the Appen-
dices; the above references provide insight on global validations. Ship SST and wind speed data
sliced in 10 km intervals describe the cross-shelf gradient in Figure A1. The coarser products (cf.
Table 1) under-represent the inshore upwelling, and are thus restricted to large-scale winds and
rainfall. Annual cycle inter-comparisons of index-area SST and zonal wind 1980-2015 are given in
Figure A2, and suggest that model seasonality is ~10% greater than observed.
**Results**
**Study area and large-scale trend maps**
The study area is illustrated in Fig 1a, and shows steep topographic and bathymetric gradi-
ents, with >1000 m mountains in latitudes < 33ºS, the coast at 34ºS, shelf-edge at 35ºS and deep
ocean to the south. The coastline is convex and indented by two bays and associated capes; the con-
tinental slope steepens eastward. The vegetation trend map (Fig 1b) reflects a warming rate of
.1ºC/yr since 1980 that increases northwest inland in conjunction with potential evaporation losses
(-.005 mm day$^{-1}$/yr). Coastal cities of Port Elizabeth and East London have slower rates of warm-
ing. The coarse-scale ECMWF-20c trend maps for SST and zonal wind (Fig 1c,d) reveal a warming
.02ºC/yr in the Agulhas Current retroflection and reduced values in the sub-tropical zones, con-
sistent with Dlomo (2014), where easterly winds are accelerating 1900-2010 (U = -.01 m s$^{-1}$/yr).
Easterly winds have accelerated in the south Atlantic and south Indian anticyclones and over the
interior of southern Africa, but in the southern mid-latitudes a westerly trend is noted over the 20[th]
century.
The ECMWF-20c trend map for precipitation minus evaporation (Fig 1e) indicates a grow-
ing deficit in the Mozambique Channel, the source region of the Agulhas Current (Fig 1f). Weaker



deficits are noted over the South Atlantic, while weak surplus trends are found over the eastern
highlands of South Africa and in the South Indian Ocean mid-latitudes. The shelf-edge Agulhas
Current converges and accelerates just east of the study area, then fans out and retroflects (Lutje-
harms 2006).

**Regional ocean trend maps and sections**

The NOAA SST trend map shows warming > .05°C/yr along the shelf edge 1981-2016 (Fig

2a) similar to Rouault et al. (2010). Yet inshore there is a distinct cooling trend that is faster in Al-
goa Bay than elsewhere (-.03°C/yr). Trends in SODA-3 salinity are weakly positive along the coast
in the period 1980-2015, suggesting reduced river run-off and greater evaporation. Surface layer
flow is accelerating in the shelf-edge Agulhas Current, particularly downstream from the study area
(Fig 2c). Outside the current, a pair of gyres (36°S, 25° & 29°E) directs flow toward the coast. This
onshore pattern has little context and may be set aside until confirmed elsewhere.

SODA-3 depth section trends (Fig 2d,e,f) show that the warming trend at the shelf edge is

aligned with an accelerating Agulhas Current (U = -.006 m s$^{-1}$/yr at 35.3°S; Backeberg et al. 2012).
The cooling trend along the coast is confined to a shallow layer < 40 m. Trends in the meridional
circulation reveal upwelling at depth and offshore transport in the near-shore zone. There is a sharp
transition to downwelling and onshore transport seaward of 35.6°S. Taken together the trend is for
convergence onto the Agulhas Current and faster downstream advection at the shelf edge. Changes
in the Agulhas Current exhibit little vertical shear, consequently cyclonic vorticity-induced uplift is
uniformly available but concentrated by the shelf slope (Lutjeharms 2006).

**Regional wind and pressure trends**

Trend maps are illustrated for reanalysis winds and latent heat flux in Fig 3a,b. Winds show

a distinct shift toward easterly winds 1980-2015, linking the South Atlantic and South Indian anti-
cyclones. The wind trends follow the convex coastline and divide zones of rising and falling latent
heat flux, consistent with the SST trends (cf. Fig 2a). The reduced moisture flux in the terrestrial




environment promotes hydrological deficit.

Regional atmospheric circulation trends were studied via EOF analysis of Dec-Feb daily

SLP data. This helps place the transient weather into long-term context. Mode-1 accounts for 38%
of variance (Fig 3c,d). Its loading pattern shows a mid-latitude antityclone passing eastward over a
5-day period, followed by a trough along the west coast that subsequently spawns a coastal low.
The mode-1 time score shows fluctuations within an upward trend (slope = .008 hPa/yr, $r^2$ = 11%),
indicating more frequent anticyclonic ridging. The gradual poleward shift of the subtropical wind
belt is comprized of pulsed synoptic weather.

**Shelf analysis and gradients**

Hovmoller plots were constructed across the southern shelf (Fig 4a,b) for 18-month filtered

surface temperature and zonal currents. Warm spells during westerly wind-driven downwelling
contrast with cool spells during easterly wind-driven upwelling: a multi-year alternation modulated
by Pacific El Nino / La Nina (Jury 2015, 2019) and the Southern Annular Mode (Malan et al.
2019). Yet there is a background trend toward inshore cooling and offshore warming that intensifies
the coastal gradient. The hovmoller plot of SODA-3 zonal currents (Fig 4b) reveals a 'pulsed' in-
tensification and coastward shift, contributing to near-shore uplift > 4 m/day (34.1-34.4°S). Thus
current- and wind-induced upwelling become additive.

Index-area time series of reanalysis and projected zonal currents (Fig 4c) show an accelerat-

ing tendency. Past and future linear regression slopes are -.0076 m $s^{-1}$/yr, with trend correlations
rising from -.81 to -.90. Future ($2^{nd}$ order) trends overlie those from past reanalysis and year-to-year
fluctuations are consistent despite technology artifacts of satellite altimetry and ensemble averag-
ing. Appendix A2 compares the index-area annual cycle of model vs observation.

The trend of NOAA SST analyzed in coastal (33.9-33.8°S) and shelf-edge (35.0-34.9°S) lat-

itudes show contrasting values but little change over the annual cycle in Fig 5a. Shelf-edge waters
are warming steadily (r= +.5) while coastal waters are cooling (r= -.5), slightly moreso from Febru-





ary to May (slope -.04°C/yr). Together these indicate a tightening gradient ($\partial T/\partial y$) and a steepening
sea slope. The annual cycle of index-area zonal wind trends (Fig 5b), averaged over three reanal-
yses, reveals that easterly winds are intensifying during summer (Nov-Feb), when subtropical ridg-
ing is most likely.
Regression of SST and winds onto the southern oscillation index (Fig 5c,d) reveals trend
patterns similar to climate change: inshore cooling (mainly summer) and offshore warming (all-
year). Winds with respect to high-phase SOI are from northeasterly and considerably stronger in
summer, hence wind-driven coastal upwelling is favoured during La Nina. The southern oscillation
index has shown an upward trend during the satellite era, and its regression onto regional sea level
air pressure patterns (cf. Fig A3) matches the earlier mode-1 pattern of mid-latitude high / sub-
tropical low (cf. Fig 3c). Long-term and multi-decadal trends are acknowledged to be additive here.
**Hydrology trends**
The earlier discovery of increasing near-shore salinity was related to drying trends in the ad-
jacent terrestrial climate, supported by declining latent heat flux (cf. Fig 3b). In Fig 5e the regional
hydrology is studied using the combined Gamtoos and Sundays River discharge record. Although
flood / drought events and 2-5 yr cycles are evident, there is no appreciable trend. The study area
lies between a zone of reduced cloudiness (Benguela – Namib) to the northwest and increased
cloudiness to the southeast, as seen in the trend map for satellite net OLR (Fig 5f). Increasing salini-
ty along the south coast (cf. Fig 2b) may be ascribed to advection from the Mozambique Channel,
where evaporation exceeds precipitation (cf. Fig 1e). Vertical motions over the shelf could also play
a role (cf. Fig 2f), whereby cyclonic shear lifts salty water.
**Model projections under greenhouse warming**
Spatial maps of ECWMF-esm rcp8.5 trends for zonal wind and rainfall 1980-2100 show a
key feature southeast of the study area (Fig 6a,b). Easterly winds are projected to increase and
rainfall is expected to decrease. The warm moist air carried westward beneath a stable inversion





layer generates less evaporation, so rain-bearing storms are projected to diminish in strength and
be deflected poleward by the sub-tropical anticyclone.

Time series of index-area values comparing ECMWF-20c reanalysis with ECMWF-esm

and Hadley-esm projections are given in Fig 6c-f. Coupled ensemble values overlie the observa-
tion-based product indicating little bias but lower variance. Zonal winds that oscillate in a station-
ary manner through the 20[th] century tend toward easterly (-U) in conjunction with declining pre-
cipitation. Air temperatures show a gradual rise during the 20[th] century in both reanalysis and
overlapping simulation. Thereafter, the warming trend steepens due to the greenhouse scenario.
There appears to be little moderating influence of cooler nearshore SST, which coarse resolution
products under-represent (cf. Fig 1c). The SOI time series is relatively stationary, but larger am-
plitude swings are noted in the early 20[th] and late 21[st] century. High phase (Pacific La Nina)
events seem steady but El Nino events appear to deepen after 2040. In summary, past zonal winds
of 1 m/s (after cancellation of east-west components) are projected to reach -1 m/s by 2050. Past
rainfall of 1.5 mm/day declines below 1 mm/day, and air temperatures of 17°C rise above 20°C
by 2050. The regression $r^2$ fit of trends are in the range from 72-97% and suggest sustained
changes for temperature, however wind and rain tend to oscillate in the ECMWF-esm projection
until the rcp8.5 scenario prevails.

In addition to ENSO influence, the Southern Annular Mode (SAM) plays a role in the lati-

tude and intensity of basin-scale anticyclonic gyres that support the Agulhas Current (Yang et al.
2016; Elipot and Beal 2018). The long-term trend in the SAM is a contraction of circumpolar
westerlies that enables poleward expansion of the tropical Hadley circulation and belt of easterly
winds rounding the tip of Africa seen here (cf. Fig 6a). Yet SAM trends are flattening with recov-
ery of the Antarctic ozone hole (Arblaster et al. 2011), and may exert less effect in future.
**Discussion and summary**

This study addressed a range of questions around spatial patterns in trends and uncovered

evidence of a pulsed poleward shift of the subtropical ridge (cf. Fig 3c,d). Analysis of land-



atmosphere-ocean conditions revealed intensified coastal upwelling from a faster shelf-edge current
and increased easterly wind, with consequences for coastal dessication. Employing coupled reanal-
ysis and model projections to distinguish coast and offshore features, a unifying process was found:
summer-time wind-driven upwelling enhances geostrophic gradients and the Agulhas Current. The
technology is reaching consensus, yet interpretations need not favour one process over another:
wind vs current, fluxes vs advection, multi-decadal vs trend, local vs remote. We do not expect one-
dimensional answers.

To place these results in context, trends in coastal SST were analyzed around the world. All

of the upwelling zones show cooling < -.03°C/yr over the satellite era: SW Africa (Namibia) 35-
20S, NW Africa (Sahara) 15-30N, SW America (Peru) 5-25S, NW America (California) 30-40N,
NE Africa (Somalia) 10-15N, and even NE America (Carolinas) 30-40N. Only the Carolinas have a
warm current offshore – like the south coast of South Africa, whereby the intensification of inshore
gradients could produce faster shelf-edge flow. However the Gulf Stream is decelerating (Jury
2020), unlike the Agulhas Current. Figure 4c gave evidence of a significant increase in westward
currents off Cape St Francis (35.5-33.5°S, 24.5-26.5°E) using low resolution ocean reanalysis and
coupled model projections. Yet some studies have found that wind-driven eddies are broadening the
Agulhas Current and that multi-year fluctuations prevail over long-term trends (Elipot and Beal
2018). International monitoring efforts such as the ASCA line (Morris et al. 2017) could resolve
ambiguities arising from the extrapolation of short-term records.

Another way of placing these results in perspective is to compare trends in coastal SST with

variance from the annual cycle (i), inter-annual variability (ii), and intra-seasonal fluctuations (iii).
The index-area standard deviations are: 2.5°C (i), 0.7°C (ii), and 0.9°C (iii) respectively, compared
with a 35-yr decline in coastal SST of -2.4°C. Applying linear regression to coastal SST data with
and without the annual cycle achieves r = -.29 vs -.76. Either way the trend is significant, not only
statistically but in terms of environmental impact.



In this study, modern reanalysis datasets have been used for mapping the marine climate
trends over the southern shelf of South Africa. Cross-shelf gradients in sea temperatures, latent heat
flux, currents and upwelling are apparent in the satellite era. SST in the offshore zone are warming
(.05°C/yr) since 1980 and there is a trend toward easterly winds, mainly in summer (U = -.015 m s$^-$
$^1$/yr). The shelf-edge Agulhas Current is accelerating (U = -.006 m s$^{-1}$ /yr) due to winds over the
west Indian Ocean (Backeberg et al. 2012) that align with the local forcing seen here. The faster
current and 'following' wind induces coastal uplift (Leber et al. 2017) and cooling (-.03°C/yr). As
the sub-tropical ridge is drawn poleward, the cross-shore gradient steepens (cf. Fig A1). Cooler
near-shore sea temperatures contribute to atmospheric subsidence, drying trends (-.005 mm day$^-$
$^1$/yr) and vegetation warming (.1°C/yr). Similar trends in local air-sea interactions are attributed to
more frequent wind-driven coastal upwelling in Malan et al. (2019). Coupled ensemble projections
from the Hadley and European models indicate that the shift toward drier weather, easterly winds,
coastal upwelling and a faster Agulhas Current may be sustained through the 21$^{st}$ century. Some of
the environmental changes could benefit marine productivity and create opportunities for resource
adaptation. Likely socio-economic consequences include an enhanced fishery that could spark in-
terest in aquaculture and ecotourism.
While the shelf may benefit, terrestrial water resources could be headed towards greater
stress. Although the hydrology is transitionally located between a drying west and moistening east,
the Sundays River sees inter-basin transfers while the Gamtoos River depends on agricultural 're-
cycling'. In both cases reduced runoff linked to rainfall could inhibit freshwater fluxes to the coastal
ocean (cf. Fig 6b).
Parallel work on this geographic niche (Jury 2019, Jury and Goschen 2020) is on-going and
further studies will: i) compare observation and reanalysis trends, ii) consider how changing satel-
lite technology represents shelf dynamics, iii) quantify wind- vs current-driven upwelling, and iv)
analyze coupled models capable of detecting sharp coastal gradients.



## Acknowledgements

SAPSE funding support from South Africa is acknowledged. Most data derive from websites of the

IRI Climate Library, KNMI Climate Explorer and Univ Hawaii APDRC.

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





**Table 1** Datasets used in the analysis, web sources are listed in acknowledgement.

| ACRONYM | NAME | RESOLUTION | TIME SPAN |
|---|---|---|---|
| CFSr-2 | Coupled Forecast System v2 ocean and atmosphere reanalysis | 0.5 deg | 1980-2015 |
| ECMWF-int | European Community Medium-range Weather Forecasts interim atmosphere reanalysis | 0.5 deg | 1980-2016 |
| ECMWF-20c | European Community Medium-range Weather Forecasts 20th century atmosphere reanalysis | 1.0 deg | 1900-2010 |
| ECMWF-ora4 | European Community Medium-range Weather Forecasts ocean reanalysis | 1.0 deg | 1958-2016 |
| ECMWF-esm | European Community Medium-range Weather Forecasts coupled ensemble 21st century projections | 1.2 deg | 1980-2100 |
| Hadley-esm | Hadley Centre coupled ensemble model 21st century projections for oceanography | 1.5 deg | 2005-2100 |
| MERRA-2 | Modern Era Reanalysis for Research and Applications v2 (NASA) | 0.5 deg | 1980-2015 |
| NOAA | National Oceanic and Atmospheric Administration surface temperature and net outgoing longwave radiation | 0.25-1.0 deg | 1980-2016 |
| SADCO SADW | S. Africa Data Centre Oceanogr. S.A. Dept. of Water Affairs | In-situ measurements | 1950-2015 1980-2016 |
| SODA-3 | Simple Ocean Data Assimilation Reanalysis v3 | 0.5 deg | 1980-2015 |




**Figures**

Figure 1 (a) Topography / bathymetry of the study area with index for temporal analyses (box) and Port
Elizabeth (dot). (b) Linear trend in annual NOAA vegetation temperature (C/yr 1981-2016). Large-scale
trends in annual: (c) Hadley SST (C/yr 1900-2016), (d) ECMWF-20c zonal wind (m s$^{-1}$/yr 1900-2010),
with inner study domains, and (e) ECMWF-20c precipitation minus evaporation trend (mm day$^{-1}$/yr). (f)
SODA3 mean 1-100 m currents (vector). Geographical labels are given in (b,c).







Figure 2 Regional trends in annual (a) NOAA sea surface temperature (C/yr 1981-2016); SODA-3 1980-2015: (b) 1-10 m salinity (ppt/yr) with section denoted, and (c) 1-50 m currents (m s$^{-1}$/yr vector); and depth section averaged 24.5-26.5E of (d) sea temperature (C/yr), (e) zonal current (m s$^{-1}$/yr), and (f) meridional circulation (m s$^{-1}$/yr vector, with W exaggerated).




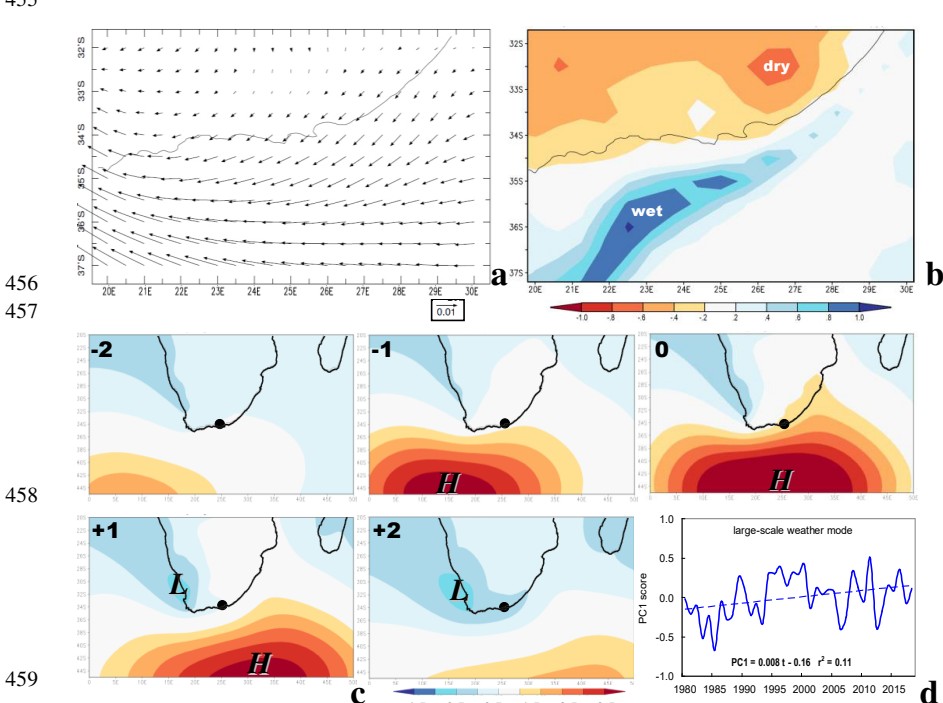




Figure 3  (a) Regional trend in annual CFSr-2 surface wind (m s$^{-1}$/yr vector, 1980-2015) and (b) latent heat
flux (W m$^{-2}$/yr). (c) Large-scale summer weather mode-1 in daily ECMWF sea level air pressure principal
component loading pattern at lags -2, -1, 0, +1, +2 days (hPa) and (d) time score. PC1 represents 38% of
variance, dot in (c) is the study area, inset in (d) is the slope and fit of the linear regression.


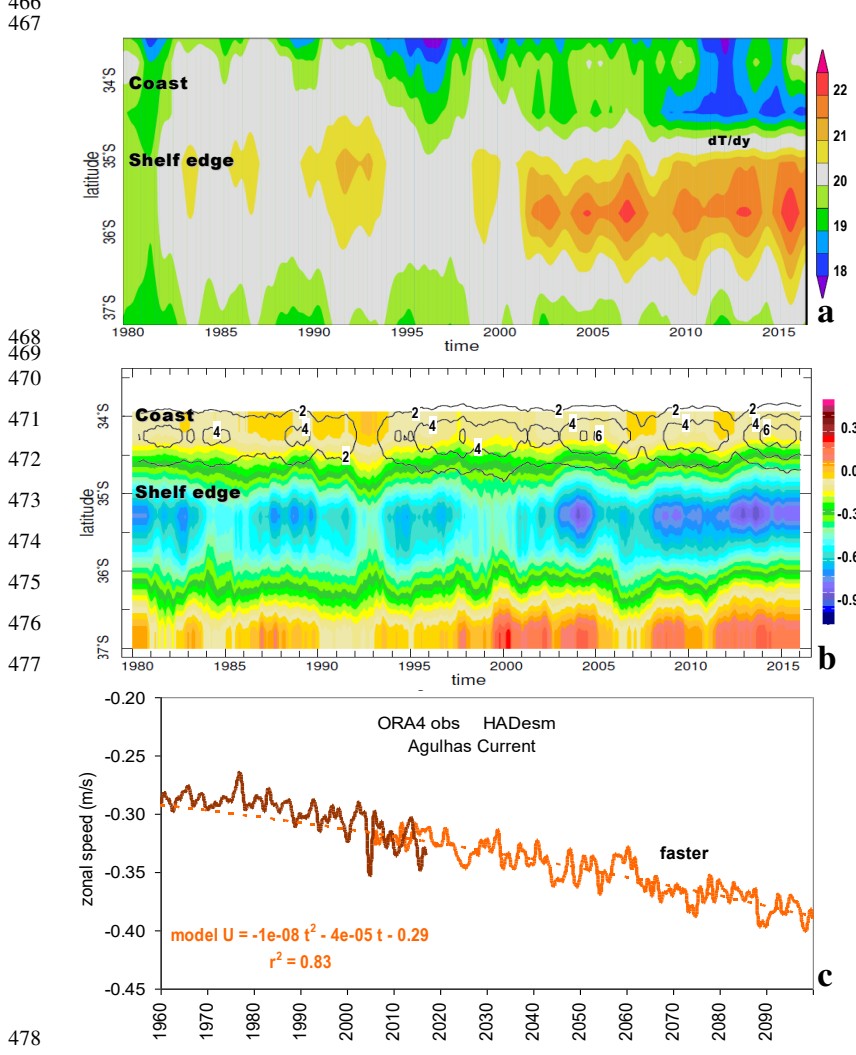


Figure 4  Hovmoller plots of 18-month filtered variables averaged 25-26E of ECMWF reanalysis: (a) sur-
face temperature (C), (b) SODA-3 1-50 m zonal current (shaded m/s) and 1-200 m upward motion (con-
tour, m/day), with coast and shelf edge denoted. (c) Index-area time series of ECMWF-ora4 observed and
HAD-esm projected 1-50 m zonal current.



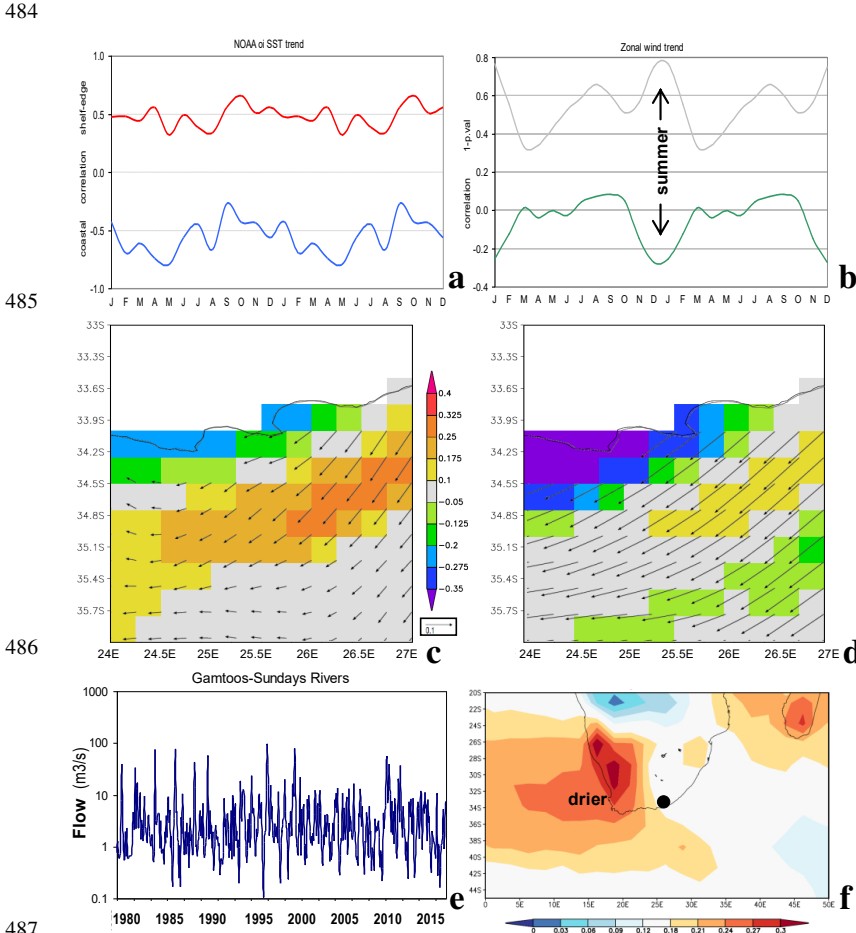




Figure 5 Analysis of monthly index-area trends for (a) coastal and shelf-edge SST, and (b) zonal wind and

its significance (1-p value), with 35 degrees of freedom. Regression of (c) annual and (d) summer NOAA

SST (shading) and SODA-3 surface wind (vector) with the SOI index 1981-2016 (units are °C and m/s per

SOI fraction). (e) Observed discharge of the combined Gamtoos and Sundays Rivers. (f) Trend of NOAA

net outgoing longwave radiation as a proxy for cloudiness (W m$^{-2}$/yr 1979-2017) with dot showing river

gauges.





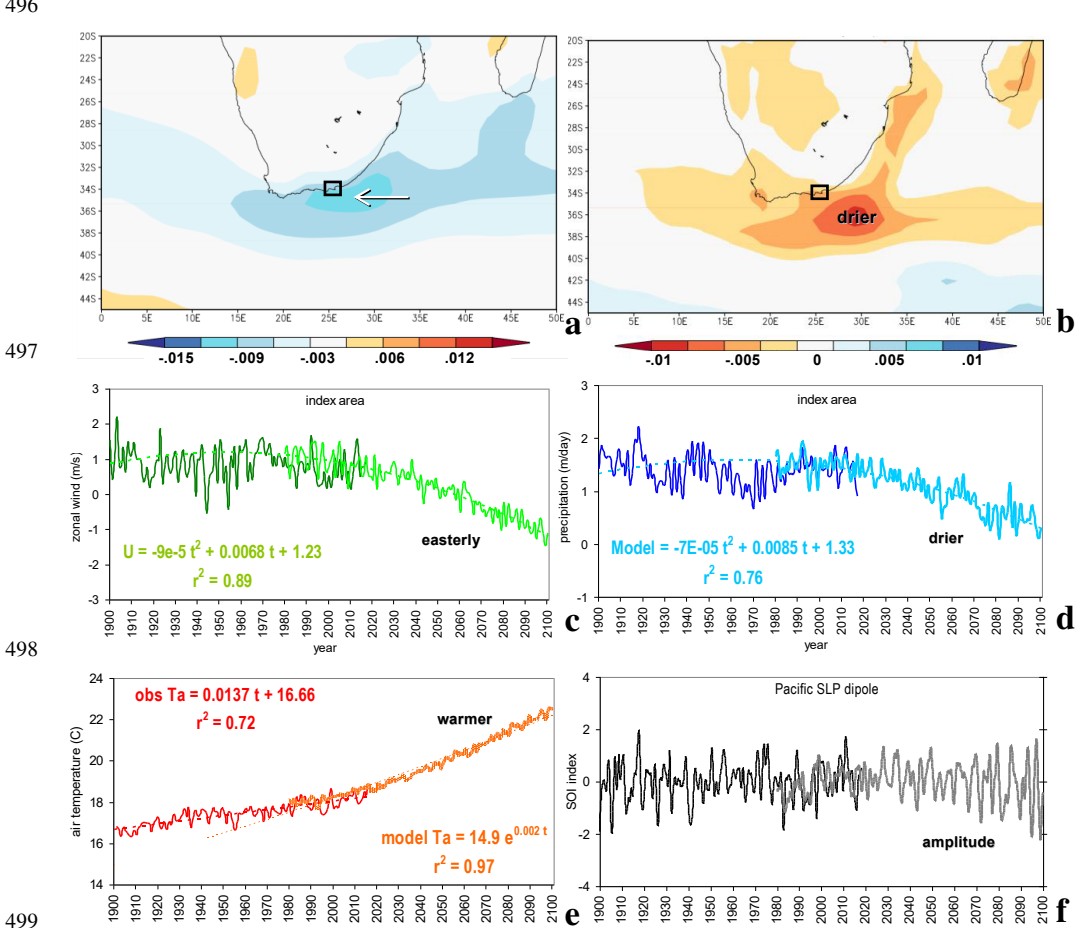




Figure 6  EC-esm projected trend maps 1980-2100: (a) zonal wind (m s$^{-1}$/yr), (b) precipitation (mm day$^{-1}$/yr).

Temporal record of index area ECMWF-20C reanalysis 1900-2010 and EC-esm projected 1980-2100: (c)

zonal wind, (d) precipitation, and (e) air temperature. (f) Observed and model projected Pacific southern

oscillation index (east-west SLP EOF mode). Best-fit trends are given; time series are composed of annual

averages.

505

506

# Appendix

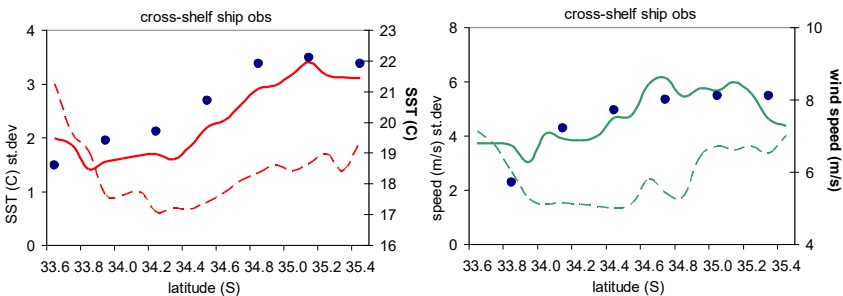

Fig A1  SADCO ship data, averages in each 0.1 latitude bin over 24.5-26.5E longitude 1950-2015; left axis

and dashed line refer to standard deviation; and comparison with 0.3 binned CFSr2 (left) and ECMWF (dots).

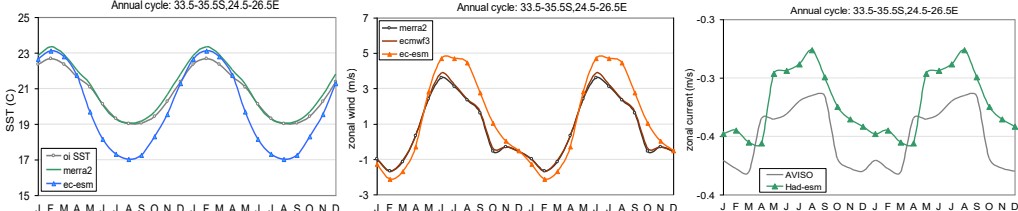

Fig A2  Annual cycles averaged over the index area; comparing model SST, zonal wind (middle) and current

with reference product. The model has an amplified annual cycle that is cooler and more westerly in winter.

Currents show summer / winter regimes with model slightly weaker and delayed.

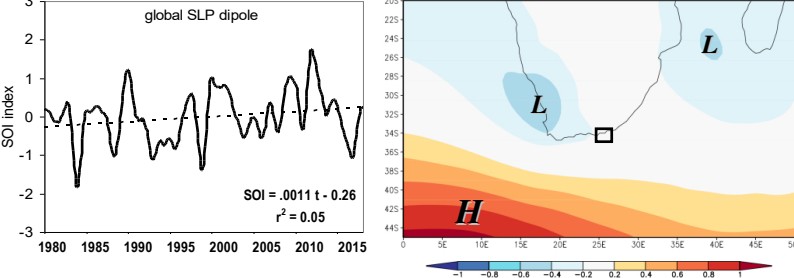

Fig A3  Graph of 18-month filtered southern oscillation index and its trend in the satellite era, and regres-

sion of Dec-Feb SOI onto regional sea level air pressure (hPa), with boxed index area.