# Peer review of "Marine climate change over the eastern Agulhas Bank of South Africa"

_Ocean Science, 2020_

## Referee Comment (RC1) · Anonymous Referee #1 · 5 Sep 2020

Evaluation Scientific significance: Good (2) Scientific quality: Fair (3) Presentation quality: Fair (3)

I- General view:

The research, in general, touches on multiple important and interconnected aspects such as oceanic, atmospheric, hydrology, and vegetation which are less studied in unison within the Agulhas Current region.

The research is based on a suite of reanalysis data sets and model projections. The manuscript sought to evaluate the oceanic and atmospheric trends, and their consecutive feedbacks subsequently establishing future projections. The manuscript also suggests a mechanistic explanation of the observed increasing rates. Overall, the research

unpacks the existent changes, their forcings, and future scenarios.

In my opinion, the subject is of interest. I do value the importance and challenges presented by this type of research for the region. This kind of research is worthy of publication in the Ocean Science (OS) journal.

Besides the scientific value of this manuscript, I suggest the authors focus their efforts in addressing the following three (broad) aspects which I will detail later in this same document (pdf file).

After reading the manuscript carefully, I suggest minor revisions. In the revised version, it should be expected that the author should; (1) appropriately introduces the topic, (2) strengthens the interpretations supporting the results and the discussion part of the manuscript and, (3) add a few simple diagnostics to solidify the mechanisms related to the dynamic of upwelling, either by following some of the suggestions below or some other that they can find on their own.

II- Comments:

- The reader expects to see three main points which are missing or less elaborated in the introduction: (1) The title for the paper implies that the focus of the paper is "Marine climate change". The author has not specified what this means, especially in the context of the east Agulhas coast of South Africa. (2) The introduction does not really introduce what the paper is about. Restructuring the literature review will improve the understanding of the study context. I will emphasize on elaborating the mechanisms of coupled air-sea interaction in the literature, and more importantly state clearly connections such as thermal feedback and current feedback in the existing previous works, and how they can impact the ecosystem accordingly. (3) What do we know about the oceanic and atmospheric trend(s) in the region, and their projections? I think that identifying these and stating the gap in the literature would elevate the importance of this work.

- Line 172: "Changes in the Agulhas Current exhibit little vertical shear, consequently cyclonic vorticity-induced uplift is uniformly available but concentrated by the shelf-slope (Lutjeharms 2006)." I would avoid such statement unless it was mechanically demonstrated, perhaps re-phrase or remove.

- Line 192 - 195: "The hovmoller plot of SODA-3 zonal currents (Fig 4b) reveals a 'pulsed' intensification and coastward shift, contributing to near-shore uplift > 4 m/day (34.1-34.4°S). Thus current- and wind-induced upwelling become additive." Based on Figure 4a,b, the author claims to have found the evidence of the upwelling mechanism based on the observational approach. I do understand the challenge that it requires to unpack the dynamic of this upwelling. Therefore, I would suggest that the upward motion indicated in Figure 4b (contours) should be replicated into Figure 4a for highlighting the correspondence between temperature upwelling signature and the upward motion. Perhaps adding a supplementary subplot will solidify this finding. I would suggest a latitude-time plot (hovmuller) of the mixte layer depth thickness (existent in Soda3) or the isopycnal slope just above the thermocline. This will inform about the dynamic of the subsurface in an upwelling event, due to the wind and/or current. The contour of the vertical motion should also be overlapped on this new subplot.

- In the context of global warming, the flux of western boundary current are already expected to increase and wind stress curl enhancing gyres are also expected to intensify. Line 225: "... revealed intensified coastal upwelling ..." implicating the Line 17-18 and Line 286.

This is a great result and a big statement that should be emphasized. The upwelling is such indicator of enhancement of these external forcings in the region (wind and current).

While this statement holds qualitatively by eye in Figure2, it would be more desirable to have more supporting results. I would expect "time series" (with linear trend) to illustrate quantitatively parameters indicating uwelling such as temperature (gradient of

temperature), density (since you use SODA), perhaps stratification. Plus, time series of wind stress in your specified box and the volume transport of the Agulhas Current. I think that this kind of diagnostic will add more conviction for this great result.

- Line 280-290: Please help the reader to clearly understand the covariance of the listed parameters in a short paragraph like a summary.

- Line 290 - 295: the discussion about the projection deserves a full independent paragraph, and should be elaborated.

III- Minor points:

- Please compute the trend per decade or per century. For climate dynamic and ocean scale point of view, trend per year sounds odd for me and less realistic (C per 10 years (decade) or C per 100 (century).

- Kindly specify where did you get these sets of data, perhaps you can insert them into your table 1.

- Lines 47 and 48, the author writes "Understanding trends in climate can inform resource management decisions and aid socio-economic uptake of..." Can the author expand on what kind of resource management decisions and socio-economic uptake they are referring to here.

- Line 159: "warming > 0.05 C/yr" (inferior or superior). This is a bit confusing for the reader. It should be fully written down and for the whole manuscript.

- Line 220: Do you mean: "Namibia"?

- Line 262, the author writes "trends in coastal SST were analyzed around the world". Please provide references.

IV- Figures:

- Please make the effort to describe properly figures in their captions.

- Figure 1f; Figure 2c,f; Figure 3a; Figure 5c,d: Arrows units (length) are not clear. I suggest overlapping arrows on top of maps with their colorbars for clarity.

V- Acknowledgement:

- It's very important to acknowledge the data sources used in this research.

Please also note the supplement to this comment:
https://os.copernicus.org/preprints/os-2020-44/os-2020-44-RC1-supplement.pdf

---

## Referee Comment (RC2) · Anonymous Referee #2 · 6 Sep 2020

I recommend against publication of this paper in the present form. The author want to solve too many questions at once which lead to serious shortcut. The narrative is interesting although the style is too telegraphic, but hypothesis are not substantiated properly and there are serious problem in the papers, the data used and the figure presented by the author do not substantiate the hypothesis and sometimes contradict his own narrative. There are a lot of problem when one look deeper in each of the issues, each of the figures or when one study the bibliography of the various topics or regional feature in details, many having been omitted. Many important papers are not cited or cited for the wrong reason or not cited for the main results. The overarching problem is that the author is trying to solve too many complex problems using web sites only which limit his ability to in depth analysis and. He is using a great number of

satellite data, model or reanalysis which are not validated for trends especially as such a small regional scale. To do a proper job would need far more figures and a complete state of the art bibliography as well as calculation not permitted by web site. I will now go to four or five major issues raised by the paper and I suggest 3 possible papers with more in depth study and analysis and up to date bibliography especially that the authors conclude by giving a future scenario for the relatively small region he study and this has serious social economical consequence id one would follow Jury scenario for the future.

The first issues is that the warming in the Agulhas Current and coastal cooling in the Port Elizabeth Port Alfred region as well as the possible causes, namely the intensification of the Agulhas Current has been already presented by Rouault et al. (2009) paper but Rouault (2010) a follow up paper is cited by Jury. To validate Rouault et al (2009, 2010) and Jury hypothesis would be a valuable contribution, but one would need to go a bit further. However, the main concern is that Rouault et al. (2009) and therefore Jury paper hypothesis had been seriously challenged and contradicted by many authors. Biastoch et al 2009 who made the cover page of Nature with his story made the hypothesis that the current is slowing down since the 80's , not accelerating. Beal et al. in another Nature paper has claimed that the Current is widening and not intensifying. It would be interesting to follow on Beal approach using Beal, a long time series of the transport of the Agulhas Current to look at Jury hypothesis and validate the model he is using. Model are notoriously bad when it comes to reproduces the basic annual cycle of the Agulhas Current (Hutchinson et al, 2018) never mind Interannual variability, decadal variability or trend. Most of those studies use the Agulhas transport and not the speed. That can be calculated with model and observation and July should also calculate the transport of his model. Other papers have put together that the warming is due to stronger Westerly to the South (Durgadoo et al;, Loveday et al). Anyway, it could be that the cooling inshore is due to the intensification of the upwelling dynamic created by the Agulhas Currents as stated by the authors and Rouault et al (2090) but there are a lots of problem in Jury paper regarding the increased upwelling due to increase wind. A major problem in the paper is that the dynamic upwelling created by the Agulhas Current is not found in Algoa Bay as Jury claim but in the Port Alfred upwelling further East and the dynamic is much more complex that described here, hence a paper mor focused on that issue with the proper dynamic is needed. . Another problem is that is more the meandering of the Current that its speed that is creating this sporadic upwelling which would give credence to Beal hypothesis of a more meandering and turbulent Agulhas Current.

Rouault, M., Penven, P. and Pohl, B., 2009. Warming in the Agulhas Current system since the 1980's. Geophysical Research Letters, 36(12).

Biastoch, A., Böning, C.W., Schwarzkopf, F.U. and Lutjeharms, J.R.E., 2009. Increase in Agulhas leakage due to poleward shift of Southern Hemisphere westerlies. Nature, 462(7272), pp.495-498.

Beal, Lisa M., and Shane Elipot. "Broadening not strengthening of the Agulhas Current since the early 1990s." Nature 540, no. 7634 (2016): 570-573.

Durgadoo, J.V., Loveday, B.R., Reason, C.J., Penven, P. and Biastoch, A., 2013. Agulhas leakage predominantly responds to the Southern Hemisphere westerlies. Journal of Physical Oceanography, 43(10), pp.2113-2131.

K Hutchinson, LM Beal, P Penven, I Ansorge, J Hermes 2018 Seasonal Phasing of Agulhas Current Transport Tied to a Baroclinic Adjustment of Near‐Field Winds Journal of Geophysical Research: Oceans 123 (10), 7067-7083

Concerning the wind driven coastal upwelling and potential cooling is the second serious issues. That region is not a typical upwelling region such as the South Benguela to the West or other Eastern boundary currents. In fact the plot shown by the author (Figure A1 and A2) show that the easterly component upwelling favourable is very weak and very seasonal and the wind is not upwelling favourable all year long. Its seasonal trend does not correspond to the seasonal trend in SST shown at the coast by Jury.

Moreover, the annual cycle of SST shown by Jury show warmer water in early summer when the easterly upwelling favourable is stronger and stronger trend when the (weak) upwelling season is over. This does not seem to indicate upwelling as a major driver of SST there or this indicate a serious problem in the data. The SST used is using interpolation to replace missing data and could be the source of the problem. Another problem is the SST used is an operational problem that has not been reanalysed and the methods used to produce the SST have changed leading to potential problem in trend detection. The data is using AVHHRR data, a product that does not see through clouds too. In any case the trend in SST shown by Jury or Rouault (2009, 2010) is stronger in austral fall when the wind is weak or westerly and not upwelling favourable. This contradiction can be found in the Figure presented by Jury. This still leave the possible intensification of the Agulhas Current as a driver of the coastal SST, but the warming of the Agulhas Current warming is all year long and the costal trend is seasonal.

Another related issue is that the model trend at the coast will be slave to the wind trend so will not prove that it has happened. Reanalysis are also model based and have serious biased in the region. Reanalysis are not reliable for trend analysis as they ingest new product at different time leading to spurious trends. Many papers are talking about this issue especially valid at the regional scale. Jury claim that ERA Interim is a good product but Imbol Nkwinka et al have shown that it is the worst product in the region even at the seasonal climatological scale as far as surface wind and latent heat flux is concerned. So how can product who have issues at the climatological scale can be trusted for trend. Moreover, the so-called validation paper cited by Jury are not done in the region there are some serious different between latent heat flux product. A proper study would follow on Imbol Nkwinkwa et al approach of using all available reanalyses and satellite product in the region to study, validate or compare at interannual, decadal and trend of surface wind and latent heat flux in the region in the region. If the model trend has a spurious easterly trend then the model will have a spurious cool anomaly at the coast, but it would not mean the trend is real. In fact

figure A1 show that the wind reanalysis is quite different from observation by a factor of two. Figure A2 show that most of the year the wind is not upwelling favourable and that water is warmer in Summer that in winter, following the annual cycle of radiation rather than being cooled by the easterly wind. This seriously questions the paper findings. It is a very complex issue. Contrary also too what the author write in the paper the ERA interim wind is the less reliable surface wind along the coast of South Africa by and seriously underestimate the mean annual cycle of wind speed. Note that the Agulhas Current is not forced by local wind so the local increase in wind speed cannot be responsible for the change in current but could generate more upwelling. This should be investigated and debated a depth. This would show that the dataset used were not handpicked to justify the narrative but that all venue were explored given the uncertainty in using ocean model and reanalysis or coupled model at the regional scale

Imbol Nkwinkwa N, A S, M, and J A. Johannessen. "Latent Heat Flux in the Agulhas Current." Remote Sensing 11, no. 13 (2019): 1576.

Up to date bibliography needs to be done as most recent relevant work on the problem of coupled model in the region. Dieppois et al, 2016, 2019 has shown that CMIPI5 coupled model do not represent winter rainfall at all and do not represent the impact of ENSO on the region and the region was the all Southern Africa a domain 100 time larger that Jury domain study. Validly of 20 century reanalyses is also questionable and probably only model driven in the South Hemisphere. Moreover the Agulhas Current is not represented in CMPI5 coupled model. So all results concerning coupled model, global arming scenario and trend are very questionable in view of the large literature showing the problem of coupled model to represent regional climate properly, even ENSO. Dieppois et al, 2016, 2019 has also shown that decadal variability does exist in the region with a 10 to 12-year cycle and another 20 25 cycle linked to the Pacific and that those cycle not represented in Coupled model. In fact it could well be that the trends presented by the author is linked to the decadal variability of climate and not

linked to global warming. IT seems that the trend in the Agulhas Current is weakest that it was 10 years ago which means that the trends is reversing. One would need to look at annual and seasonal mean and find out if the trend is really linear as it is more or less for global warming temperature or if it is just part of natural decadal variability As for the coupled model due to their low resolution and their problem to represent interannual variability, the effect of ENSO or decadal variability of Southern Africa climate, they cannot be used here. Unless they are validated. Paper exists showing those problems (work done by Dieppois et al, 2015, 2019, Pohl et al 2019). All in all it is very important to find out if the trend is linked to global warming trend and could carry on in the future or it is just par of natural variability. This is another paper by itself.

Another concern is that El Nino leads to weaker upwelling in the region and drought while La Nina leads to colder coastal water and more rainfall inland. This is the opposite effect as the effect proposed at the decadal scale or trends scale by Jury. This leads to a very serious concern. Why does the reduction in SST in Algoa Bay leads to the desiccation of the inland region, coastal water is not warm enough to lead to convection with potential advection of rainfall inland? However since the wind trend shown is easterly and not Southerly there is little chance that the desiccation is linked to a change in the moisture flux due to a cooled Algoa Bay. A stronger wind will lead to more evaporation too. Also the figure of satellite estimate of vegetation temperature change is not convincing as many areas have not changes and some remote region seems o be drier. Once again the dataset is using different instruments and maybe unreliable for trend analysis and it was not reanalysed for that matter. This product should be validated interannually first. A proper demonstration would involve looking the transport of moisture inland but also the generation of rainfall in more details. 100 of rainfall station are available in the region, why not used them. River data is not reliable and human abstraction can lead to decrease in the flow of those sporadic and small rivers. The Kruger paper cited regarding trend in rainfall do not show any trend in rainfall. The trend in wind speed is easterly not northerly so how does it work. What are the mechanism linking relatively cold water to rainfall inland? Where are not in a

monsoon region and only when water is > 15 C that we find an effect of sea surface temperature on rainfall. It is not even clear why the coastal ocean and Algoa bay is a driver of rainfall in the region, a region where rainfall is brought about by large scale rainfall system such as huge cyclonic system such cold front, cut off low, tropical temperate trough or inland low pressure system that create mainly a offshore and not an onshore flow. Again more need to be done to proof that hypothesis. Moreover the increase in latent heat flux on the nearby Agulhas Current should largely compensate for the small loss in latent heat flux at the coast. Latent heat flux in Western Boundary current is 3 to 5-time higher than surrounding water and Jury is presenting a very strong trend in latent heat flux nearby. If Jury hypothesis is correct , this should be presented in another paper.. Correlation is not causality. Also, the author shows that the Agulhas Temperature and associated latent heat flux have substantially increase. This should bring more moisture to the coast when the wind is onshore, and this could compensate largely for the inshore cooling. This is not investigated or discussed at all in the paper. This probably the weakest point of the paper and I recommend the author to make a third paper in the issue as it could be independent of the rest.

---

## Referee Comment (RC3) · Anonymous Referee #3 · 7 Oct 2020

General comments: The topic is of interest, and the authors do a good job considering many data sources, and doing their best to integrate these pieces together. However, the paper could be introduced better, and some revisions are necessary particularly addressing the possible strengthening of the Agulhas Current.

My major points of revision on the paper regard three issues:

1) There is not very much information on how well these reanalyses simulate the region, particularly the velocity field. Does SODA-3 really simulate the real ocean well enough that we can trust a trend of -0.006 m s-1 y-1? I am doubtful, with the information given, but maybe you can convince the reader otherwise.

2) You discuss at various points a strengthening or acceleration of the Agulhas Current.

However, you only compute averages and show figures for the near surface (example: Fig 4c shows top 50 m). A stronger velocity in the upper 50 m does not necessarily imply the Agulhas Current as a whole has strengthen. There could be compensating changes at depth. Please compute trends in Sv yr-1, and state along with your trends in upper ocean m s-1 yr-1 so that the reader can assess whether you are truly seeing a strengthening current.

2) The observational paper which assesses a possible trend in the Agulhas Current volume transport finds no strengthening since 1993, and instead a broadening of the current (Beal & Elipot, 2016). While you do briefly mention this at line 270-273, there is no comparison to your results. Why do you suspect your results are different? Is it due to deficiencies in SODA3? Or due to the specific end years in Beal & Elipot, 2016, or some other methodological difference? If you compute a trend in Sv/yr, over the full depth of the Agulhas Current, over the same time period as Beal & Elipot, what do you get? I also think it would be useful to cite and discuss this paper earlier, possibly either in the introduction or in the paragraph starting at line 166.

Specific comments:

Introduction: A stronger introduction that addresses why the reader should care would be helpful. i.e., fishing industry, weather impacts, climate feedbacks, etc. due to any changes in the region.

Line 39: The Agulhas Current meanders a mean of 1.6 times per year, add citation to Elipot & Beal (2015)

Line 80: Citation or figure to show this?

Line 82: Please give some description of what "transient EOF" is and how this varies from a general EOF

Line 94-96: It would be helpful if you illustrated these regions on one of your maps, perhaps in Figure 1.

Line 112: illustrating the regions on a map would help

Line 160: Please state value from Rouault (2010)

Line 164-165: Do you think this is real? Or are you saying it is an artifact of SODA3 that may not exist in the real ocean?

Line 191: Elipot & Beal (2018) is another useful point of comparison

Line 196: Need to make it clear you are only looking at velocity trends of upper 50 m. This is not necessarily equivalent to an acceleration of the full depth Agulhas Current.

Line 214: Not sure what you mean by "Long-term trend and multi-decadal trends are acknowledged to be additive here". Perhaps you need to explain in more detail

Line 219: State value, even if it is not different from zero considering the error

Line 258-261: I am not sure what these two sentences mean. What is the technology which is reaching consensus?

Line 262: Please show this global analysis in a figure.

Line 272-273: You may want to cite two recent papers presenting data from the ASCA array: McMonigal et al. (2020) JPO, doi: 10.1175/JPO-D-20-0018.1, Gunn et al. (2020) accepted to JPO, no doi yet.

Line 292-295: It is not clear to me how fisheries would benefit from these changes, besides the increased upwelling. But I am not sure what a faster Agulhas Current, drier weather, or increased easterly winds would do to marine productivity. Please describe and/or cite papers.

Line 302: I strongly suggest that you do more comparison of the reanalysis and observational trends within this paper. It does not need to be extensive, but compare your trends to those from previous observational papers and explain why they agree (or why they disagree).

СЗ

Figure 2: Why are panels d,e,f only shown to 200 m depth? Is there no trend below that?

Figure 3: very unclear that panel c is actually 5 panels.Maybe relabel so it is clear what c refers to

Figure 4: Please include a trend in in Sv yr-1, integrated over full depth, as well as in m s-1 yr-1 in upper 50 m. Otherwise, it is conceivable that your "accelerating" Agulhas is simply more surface intensified, but not actually strengthening.

Figure A1: What are the solid and dashed lines?

Citations: Elipot, S., & Beal, L. M. (2015). Characteristics, Energetics, and Origins of Agulhas Current Meanders and Their Limited Influence on Ring Shedding. Journal of Physical Oceanography, 45(9), 2294–2314. https://doi.org/10.1175/JPO-D-14-0254.1

Elipot, S., & Beal, L. M. (2018). Observed Agulhas Current sensitivity to interannual and long-term trend atmospheric forcings. Journal of Climate, JCLI-D-17-0597.1. https://doi.org/10.1175/JCLI-D-17-0597.1

Technical corrections: Line 97: "reduction to" should be "reduction of". Also unsure what "(Dec-Feb) interval" means.

Line 148: "accelerating 1900-2010" should be "accelerating over 1900-2010"

Line 282: "SST ... IS warming"

Figure 4c caption: Do you mean meridional current? The Agulhas Current is (mostly) oriented southwards

---

## Author Comment (AC1) · 13 Oct 2020

Feedback on Algoa marine climate change: Ocean Sci. Discuss.,
https://doi.org/10.5194/os-2020-44-RC1, 2020
**Author replies in bold.**

Reviewer#1
I- General view: The research, in general, touches on multiple important and
interconnected aspects such as oceanic, atmospheric, hydrology, and vegetation which
are less studied in unison within the Agulhas Current region. The research is based on a
suite of reanalysis data sets and model projections. The manuscript sought to evaluate the
oceanic and atmospheric trends, and their consecutive feedbacks subsequently
establishing future projections. The manuscript also suggests a mechanistic explanation
of the observed increasing rates. Overall, the research unpacks the existent changes, their
forcings, and future scenarios. In my opinion, the subject is of interest. I do value the
importance and challenges presented by this type of research for the region. This kind of
research is worthy of publication in the Ocean Science (OS) journal. Besides the
scientific value of this manuscript, I suggest the authors focus their efforts in addressing
the following three (broad) aspects, as outlined below…

In the revised version, it is expected that the author should: (1) appropriately introduce
the topic, (2) strengthen the interpretations supporting the results and the discussion part
of the manuscript and, (3) add a few simple diagnostics to solidify the mechanisms
related to the dynamic of upwelling, either by following some of the suggestions below
or some other that they can find on their own.
**Author did re-writing to extend the introduction and discussion, eg. (1) and (2)
above. On (3) above, the author added further interpretations on wind- and current-
induced upwelling.**

II- Comments: - The reader expects to see three main points which are missing or less
elaborated in the introduction:

(1) The title for the paper implies that the focus of the paper is "Marine climate change".
The author has not specified what this means, especially in the context of the east
Agulhas coast of South Africa. **Abstract says rate of change in the marine
environment, this is repeated on line 49.**

(2) The introduction does not really introduce what the paper is about. Restructuring the
literature review will improve the understanding of the study context. I [suggest]
elaborating the mechanisms of coupled air-sea interaction in the literature, and more
importantly state clearly connections such as wind-current-thermal feedback… in
previous works, and how they can impact the ecosystem accordingly. **New paragraph
was added on lines 40-48.**

(3) What do we know about the oceanic and atmospheric trends in the region, and their
projections? I think that identifying these and stating the gap in the literature would
elevate the importance of this work. **Covered by response to (2) above.**

- Line 172: "Changes in the Agulhas Current exhibit little vertical shear, consequently cyclonic vorticity-induced uplift is uniformly available but concentrated by the shelfslope (Lutjeharms 2006)." I would re-phrase such statement unless it was mechanically demonstrated… **OK, that was re-phrased with references to figure panels.**

- Line 192 - 195: "The hovmoller plot of SODA-3 zonal currents (Fig 4b) reveals a 'pulsed' intensification and coastward shift, contributing to near-shore uplift > 4 m/day (34.1-34.4∘S). Thus current- and wind-induced upwelling become additive." Based on Figure 4a,b, the author claims to have found the evidence of the upwelling mechanism based on the observational approach. I do understand the challenge that it requires to unpack the dynamic of this upwelling. Therefore, I would suggest that the upward motion indicated in Figure 4b (contours) should be replicated into Figure 4a for highlighting the correspondence between temperature upwelling signature and the upward motion. Perhaps adding a supplementary subplot will solidify this finding. I would suggest a latitude-time plot (hovmuller) of the mixed layer depth (from Soda3) or the isopycnal slope just above the thermocline. This will inform about the dynamic of the subsurface in an upwelling event, due to the wind and/or current. The contour of the vertical motion should also be overlapped on this new subplot.
**Author instead added new zonal wind stress and rainfall as Fig 4b and c, Fig 4a is surface temp, while Fig 4d shows uplift & westward currents. Author considered MLD (below), but felt that tau X and rainfall were more significant.**

[Figure]

 - In the context of global warming, the flux of western boundary current are already expected to increase and wind stress curl enhancing gyres are also expected to intensify.

Line 252: "... revealed intensified coastal upwelling ..." implicating the Line 17-18 and Line 286. This is a great result and a big statement that should be emphasized. The upwelling is such indicator of enhancement of these external forcings in the region (wind and current). While this statement holds qualitatively by eye in Figure 2, it would be desirable to have more supporting results. I would expect "time series" (with linear trend) to illustrate quantitatively parameters indicating uwelling such as temperature (gradient of temperature), density (from SODA), perhaps stratification. Plus, time series of wind stress in your specified box and the volume transport of the Agulhas Current. I think that this kind of diagnostic will add more conviction for this great result. **Fig 3a shows the wind trend that is from easterly and curving around the convex coastline. Fig 6c shows that trends forward easterly winds are most evident in the future projection, whereas the past observed zonal wind trend is weak. These notes were added in the Discussion.**

- Line 280-290: Please help the reader to clearly understand the covariance of the listed parameters in a short paragraph like a summary. **The words 'contribute to' was changed to 'correspond with' on line 285. In line 290, sentence was added to say the co-varying features are a local response to the poleward shift of the subtropical ridge.**

- Line 290 - 295: the discussion about the projection deserves a full independent paragraph, and should be elaborated. **Author does not want to focus too much on the projections, because of weak trends in past observations.**

III- Minor points: - Please compute the trend per decade or per century. For climate dynamic and ocean scale point of view, trend per year sounds odd for me and less realistic (degC / 100 yr  or century). **Author prefers to use per year.**

- Kindly specify where did you get these sets of data, perhaps you can insert them into your table 1. **Author has listed all data sources in the acknowledgements, as mentioned in table 1 caption.**

- Lines 47 and 48, the author writes "Understanding trends in climate can inform resource management decisions and aid socio-economic uptake of. . ." Can the author expand on what kind of resource management decisions and socio-economic uptake they are referring to here. **Author is not an expert there, so changed to say that past work offers guidance…**

- Line 159: "warming > 0.05 C/yr" (inferior or superior). This is a bit confusing for the reader. It should be fully written down and for the whole manuscript. **> symbol surely means 'more than'**
- Line 220: Do you mean: "Namibia"? **this word is used in line 262 correctly.**

- Line 262, the author writes "trends in coastal SST were analyzed around the world". Please provide references.
**This was done in-house by the author… the sentence was re-written.**

IV- Figures: - Please make the effort to describe properly figures in their captions.
- Figure 1f; Figure 2c,f; Figure 3a; Figure 5c,d: Arrows units (length) are not clear. I suggest overlapping arrows on top of maps with their colorbars for clarity**. Captions were improved. Vector scales were inset (over land).**

V- Acknowledgement: - It's very important to acknowledge the data sources used in this research. **Those are specified.**

Reviewer#2
**Author has used some of the criticism and deleted the 'diatribe'.**
…the author is trying to solve too many complex problems…
**The scope of research was narrowed, and over-stated interpretations were removed: line 19 of abstract.**

…warming offshore and cooling inshore (near Port Elizabeth) linked with intensification of the Agulhas Current, was presented by Rouault et al. (2009). However this hypothesis has been challenged by many authors: (Biastoch et al 2009, Beal ? transport and meandering of the Agulhas Current, Hutchinson et al 2018, Durgadoo et al, Loveday et al)… whether the cooling inshore is locally wind-driven or upstream current-induced can not be concluded.
**Yes, it is said that both processes are involved: lines 17-18, 190-191, 258-259.**

…the annual cycle of SST …is using (NOAA oi) interpolation to replace cloudy days and could be the source of the problem. … the SODA-3 and NOAA SST trend at the coast will be slave to the wind… The reanalysis are not reliable for trend analysis as shown by Nkwinkwa N, et al. 'Latent Heat Flux in the Agulhas Current' Remote Sensing…
**Author uses the best available products, based on data assimilation and past ability to predict the climate of southern Africa. The author is confident that reported trends are robust, and avoids products with poor validation.**
…Dieppois et al, 2016, 2019 has shown that CMIP5 coupled model do not represent… the EC20C reanalyses is also questionable … decadal variability exists (eg. Pohl et al 2019).
**Yes, it is said that long-term and multi-decadal trends may be additive: line 213.**

…there is little chance that the desiccation is linked to a change in the moisture flux… the author shows that the offshore Agulhas SST and associated latent heat flux have increased. This should bring more moisture to the coast when the wind is onshore…
**Author shows that the wind is more offshore and upwelling favourable in Fig 3a, so this criticism seems unsupported. New rainfall hovmoller (Fig 4c) displays a sharp boundary between dry inshore / wet offshore climates.**

**Although much of reviewer#2 criticism seems unwarranted, there are some good points, so the author added the relevant references, and narrowed the scope of research questions: lines 50-54.**

**Author re-wrote the hydrology–salinity connection:  lines 215+.**

**Throughout the text, statements that said 'the Agulhas Current is accelerating' were modified to say 'locally' eg. on the shelf-edge between 21-28E as shown in Fig 2c. On line 272 – 'Our analysis does not claim the whole Agulhas Current is strengthening, only along the shelf-edge of the eastern Agulhas Bank.'**

Reviewer#3
General comments: The topic is of interest, and the authors do a good job considering

many data sources, and doing their best to integrate these pieces together. However, the paper could be introduced better, and some revisions are necessary particularly addressing the possible strengthening of the Agulhas Current.

My major points of revision on the paper regard three issues:

1) There is not very much information on how well these reanalyses simulate the region, particularly the velocity field. Does SODA-3 really simulate the real ocean well enough that we can trust a trend of -0.006 m s-1 y-1? I am doubtful, with the information given, but maybe you can convince the reader otherwise. **See note below**

2) You discuss at various points a strengthening or acceleration of the Agulhas Current. **The author will upload a powerpoint in supplementary data, which was used for training of oceanography staff in Port Elizabeth in Aug 2019 on coupled global data assimilation. That makes clear most data inflow is near the surface and models interpolate in space and** extrapolate **in depth. In that file and in two companion papers which have published recently, some attributes of ocean reanalysis are compared with insitu observations. It seems arbitrary to take a small portion of data going into the reanalysis as verification, when vast amounts of satellite data are used together with smart ocean models which are coupled with the atmosphere and land.** However, you only compute averages and show figures for the near surface (example: Fig 4c shows top 50 m). A stronger velocity in the upper 50 m does not necessarily imply the Agulhas Current as a whole has strengthen. There could be compensating changes at depth. Please compute trends in Sv yr-1, and state along with your trends in upper ocean m s-1 yr-1 so that the reader can assess whether you are truly seeing a strengthening current. 2) The observational paper which assesses a possible trend in the Agulhas Current volume transport finds no strengthening since 1993, and instead a broadening of the current (Beal & Elipot, 2016). While you do briefly mention this at line 270-273, there is no comparison to your results. Why do you suspect your results are different? **The author does not analyze the total volume transport, but the upper layer that is better observed.** Is it due to deficiencies in SODA3? Or due to the specific end years in Beal & Elipot, 2016, or some other methodological difference? If you compute a trend in Sv/yr, over the full depth of the Agulhas Current, over the same time period as Beal & Elipot, what do you get? I also think it would be useful to cite and discuss this paper earlier, possibly either in the introduction or in the paragraph starting at line 166. Specific comments: Introduction: A stronger introduction that addresses why the reader should care would be helpful. i.e., fishing industry, weather impacts, climate feedbacks, etc. due to any changes in the region. Line 39: The Agulhas Current meanders a mean of 1.6 times per year, add citation to Elipot & Beal (2015) Line 80: Citation or figure to show this? Line 82: Please give some description of what "transient EOF" is and how this varies from a general EOF **That was re-written to say it is the EOF evolution at lags -2 to +2 days**. Line 94-96: It would be helpful if you illustrated these regions on one of your maps, perhaps in Figure 1. Line 112: illustrating the regions on a map would help **A figure is now cited.**

Line 160: Please state value from Rouault (2010) **both the + offshore and – inshore SST trends are similar to the earlier findings, it depends on location and would be cumbersome to paraphrase.**

Line 164-165: Do you think this is real? Or are you saying it is an artifact of SODA3

that may not exist in the real ocean? **Yes it is one of the few features that the author does not have full confidence.**

Line 191: Elipot & Beal (2018) is another useful point of comparison

Line 196: Need to make it clear you are only looking at velocity trends of upper 50 m. This is not necessarily equivalent to an acceleration of the full depth Agulhas Current. **Yes that is now said, acceleration is confined in depth and in breadth (eg. localized and not necessarily up- and down-stream)**

Line 214: Not sure what you mean by "Long-term trend and multi-decadal trends are acknowledged to be additive here". Perhaps you need to explain in more detail **That statement was re-written.**

Line 219: State value, even if it is not different from zero considering the error

Line 258-261: I am not sure what these two sentences mean. What is the technology which is reaching consensus? **Reanalysis is reaching consensus based on a shared data assimilation system – as now stated.**

Line 262: Please show this global analysis in a figure. **Author believes this will be distracting, this SST trend maps is shown below. The coastal upwelling regions are small in global context.**

[Figure]

Line 272-273: You may want to cite two recent papers presenting data from the ASCA array: McMonigal et al. (2020) JPO, doi: 10.1175/JPO-D-20-0018.1, Gunn et al. (2020) accepted to JPO, no doi yet.

Line 292-295: It is not clear to me how fisheries would benefit from these changes, besides the increased upwelling. But I am not sure what a faster Agulhas Current, drier weather, or increased easterly winds would do to marine productivity. Please describe and/or cite papers. **A companion paper is cited therein. Jury, M.R., 2019, Environmental controls on marine productivity near Cape St Francis, South Africa, Ocean Science, 15, 1579–1592. ocean-sci.net/15/1579/2019/ However, it is noted that increased chlorophyll content relates to fresh intrusions, hence not a drier climate, so that was re-written.**

Line 302: I strongly suggest that you do more comparison of the reanalysis and observational trends within this paper. It does not need to be extensive, but compare your trends to those from previous observational papers and explain why they agree (or why they disagree). **Intercomparisons were added as appendix Figs A1, A2, and have been done in parallel research, that is referenced: Jury, M.R. and Goschen, W., 2020, Inter-relationships between physical ocean-atmosphere variables over the shelf south of South Africa from reanalysis products, Cont. Shelf Research, 202, doi.org/10.1016/j.csr.2020.104135.**

Figure 2: Why are panels d,e,f only shown to 200 m depth? Is there no trend below

that? **Most observations and ocean influence on climate is from the surface layer, and satellite altimetry is extrapolated with depth. Fig 2e shows that trends over the shelf are relatively uniform with depth.**

Figure 3: very unclear that panel c is actually 5 panels.Maybe relabel so it is clear what c refers to **OK, that was revised.**

Figure 4: Please include a trend in in Sv yr-1, integrated over full depth, as well as in m s-1 yr-1 in upper 50 m. Otherwise, it is conceivable that your "accelerating" Agulhas is simply more surface intensified, but not actually strengthening. **The paper was re-written to indicate that strengthening is localized, and that trends are partially related to inshore upwelling (cf. Fig 2d).**

Figure A1: What are the solid and dashed lines? **That was re-analyzed.**

Citations: Elipot, S., & Beal, L. M. (2015). Characteristics, Energetics, and Origins of Agulhas Current Meanders and Their Limited Influence on Ring Shedding. Journal of Physical Oceanography, 45(9), 2294–2314. **OK, that reference was added.**

Elipot, S., & Beal, L. M. (2018). Observed Agulhas Current sensitivity to interannual and long-term trend atmospheric forcings. Journal of Climate, JCLI-D-17-0597.1.

Technical corrections: Line 97: "reduction to" should be "reduction of". Also unsure what "(Dec-Feb) interval" means. **No, the correct meaning is 'to', but the term 'interval' was misleading and fixed.**

Line 148: "accelerating 1900-2010" should be "accelerating over 1900-2010" **OK.**

Line 282: "SST : : : IS warming" **OK.**

Figure 4c caption: Do you mean meridional current? The Agulhas Current is (mostly) oriented southwards **No, in the study area the Agulhas Current is mostly westward.**

---

## Author Comment (AC2) · 13 Oct 2020

all together...

———————————————

---

## Author Comment (AC3) · 13 Oct 2020

all together...

———————————————

---

## Author Comment (AC4) · 13 Oct 2020

**Coupled data assimilation:**
**for oceanography work over South Africa's shelf**

Observations are drawn into a coupled model by data assimilation: ocean, land, atmosphere

- < 10% of observations are in-situ, the rest come from satellite.
- Operational products use past observations, whereas reanalysis products use near-future and late-arriving data.
- Ocean & land assimilation have a generous multi-day time window, but atmospheric assimilation has a narrow cut-off (few hours).
- Observations have 'influence' according to the type, accuracy & reliability, eg. in-situ above remote, calibrated above unknown.
- At a model grid-point, the observations affect the interpolated value according to proximity.
- Incoming data are constrained to model physics, climatology, persistence & prior forecast.

Data assimilation is the technique whereby observational data are combined with output from a numerical model to produce an **optimal** estimate of the **evolving** state of the system.

**The Data Assimilation Process**

[Figure]

**observations**          **forecasts**

compare
reject
adjust

estimates of state & parameters
➡ errors in obs. & forecasts

**Why We Need Data Assimilation**

[Figure]

[Figure]

[Figure]

- range of observations
- range of techniques
- different errors
- data gaps
- quantities not measured
- quantities linked

**Methods of Data Assimilation**

- Optimal interpolation (or approx. to it)

- 3D variational method (3DVar)

- 4D variational method (4DVar)

- Kalman filter (with approximations)

**Types of Data Assimilation**

- Intermittent
- Continous

**Key data streams**

- Surface in-situ measurements: ship, buoy
- Sub-surface observations: float, XBT
- Satellite remote sensing:
  - De-clouded visible & infrared
  - Passive microwave (wide swath)
  - Active microwave (narrow swath)

Model forecast, climatology, persistence
  - Forcing from atmosphere and land models
  - Theoretical calculations by ocean model

Near-surface temp obs density

https://www.godae-oceanview.org/science/ocean-forecasting-systems/assimilation-characteristics/
http://www.marineinsitu.eu/dashboard/          http://www.emodnet-physics.eu/Map/
https://www.ecmwf.int/en/forecasts/quality-our-forecasts/monitoring-observing-system#Ocean

**Coupled Data Assimilation**

Atmosphere Observations

Fast variations

Land Obs

Slow variations

Coupled Model Assimilation

Realistic condition of the Earth System

Adapting assimilation to inputs

Ocean Observations

Multi-day coverage by active MW radiometers that provide wind fields for evaporation and Ekman transport

Remember that land-based wind data are not assimilated due to 'exposure'.

[Figure]

**Coupled data assimilation**

[Figure]

**Ocean model data interpolation**

[Figure]

well distributed
satellite obs, swaths

atmospheric forcing

land surface fields

uneven *ship / buoy
observations, float
& XBT profiles*

*Model 1st guess*

*Climatology*

*Persistence*

*Ocean Physics*

*Calculations*

*from theory*

multiple
influences

*Measurements at earlier (or later) times have less influence than recent*

**Ocean observing system**

[Figure]

Floats, buoys, ships, and other in-situ obs

**Satellite altimeter obs of geoid-corrected sea surface height anomaly**

[Figure]

STATION_TYPE from 2018-09-07 00:00:00 to 2018-09-12 12:45:00 N_OBS = 371752

Saral/AltiKa    Cryosat-2    Jason-3

5-day coverage

**SA involvement in real-time ocean monitoring is limited**

[Figure]
 • Weather reports from station / ship
• Global profiling by floats and aircraft

but

[Figure]
 • No marine stations (buoys) reporting
• Harbours tide gauges are 'quiet'

[Figure]

Why is SL offline?

SAWS: best reporting in Africa

floats

surface

aircraft

The SA marine science community is skeptical of ocean reanalysis; users do not feel confident in their outcomes, so operational research has limited influence.

**SST** from IR+MW satellite with insitu-calibration, after de-clouding over multi-day window**

Latest L4 sea-surface temperature observations from OSTIA (20180912)

[Figure]

**Overlapping satellite missions to collect essential data**

[Figure]

Passive wide-swath MW radiometer

'footprint' size:   50 km  →   25 km  →   10 km

Active MW radar altimeter

Active wide-swath MW scatterometer

Ocean reanalysis products have improving technology, and resolve the coastal gradient after 2008, with the advent of higher resolution radiometers

**Satellite vs reanalysis**

- Individual satellites have orbital limitations and aging radiometers.

- Satellite 'level-3' products are corrected for radiometer drift, atmospheric contamination (reflection, scattering, absorption), coastal contamination by land fraction within radiometer footprint.

- Level-3 products are interpolated to a grid after correction, and usually represent composites of multiple images within a sliding time window.

- SST reanalysis products blend IR and MW products, to reduce contamination: eg. GHR-MUR Level-4 since 2002.

- Ocean reanalysis uses multiple level-3 satellite products from NASA, ESA, etc, in addition to in-situ & ancillary data, model physics & recent meaurements.

- Why use single satellite products?

| Instrument | Radiometer/Orbit | Resolution | Error | Issues |
|---|---|---|---|---|
| AVHRR | IR/Polar | 1 km | 0.6°C | Clouds, aerosols |
| AATSR | IR/Polar | 1 km | 0.3°C | Clouds, sparse |
| MODIS | IR/Polar | 1 km | 0.5°C | Clouds, aerosols |
| VIIRS | IR/Polar | 1 km | 0.4°C | Clouds, aerosols |
| GOES Imager | IR/Geostationary | 6 km | 1.0°C | Clouds, aerosols |
| SEVIRI | IR/Geostationary | 6 km | 0.7°C | Clouds, aerosols |
| MT-SAT | IR/Geostationary | 6 km | 0.7°C | Clouds, aerosols |
| AVHRR | IR/Polar | 9 km | 0.4°C | Clouds, aerosols |
| AMSR-E | MW/Polar | 25 km | 0.5°C | Land, rain |
| AMSR2 | MW/Polar | 25 km | 0.5°C | Land, rain |
| WindSat | MW/Polar | 25 km | 0.5°C | Land, rain |
| TMI | MW/Equatorial | 25 km | 0.5°C | Land, rain |
| Buoys/ships | in situ | variable | ~ 0.5°C | Depth, sparse |

**DA methodology:**
**ECMWF ORA5**

- ▶ Methodology is 3D-Var–FGAT
- ▶ Assimilation of in situ profiles, SLA, SIC
- ▶ Relaxation of SST towards OSTIA
- ▶ OCEAN5 is a reanalysis-analysis system with 2 streams – behind real-time and real-time
- ▶ Assimilation window varies from 8 days to 12 days and split into two chunks
- ▶ Minimisations performed separately for sea ice and ocean components
- ▶ Atmospheric forcing comes from the HRES system

[Figure]

**A sliding window for incoming data**

[Figure]

**Incremental Update Cycle**

[Figure]

Cold start, has no prior 'knowledge'

Warm start, uses persistence to 'nudge' the result

Block run, uses climatology to 'nudge' the result

[Figure]

[Figure]

[Figure]

[Figure]

**Different time-scales for ocean/land and atmospheric (NWP) modelling**

☐ NWP forecasts have to be produced in a timely fashion

☐ Not all ocean observations are available for current atmospheric cut-off times

▶ Would like coupled assimilation for:

- ☐ Coupled observation operators
- ☐ Atmospheric bias correction of ocean sensitive satellite observations
- ☐ More balanced initial conditions

▶ Outer loop coupling with the atmosphere – lots of potential to help with bias correction and screening of ocean sensitive satellite observations

- ▶ Aligning the ocean analysis window to the current atmospheric window would mean missing lots of vital in situ observations
- ▶ Care needs to be taken not to inherit ocean model biases into the atmospheric analysis

**Atmospheric Model**

- Aerosol processes (Microphysics)
  - Nucleation/condensation
  - Phase changes
- Cloud processes
  - Conden./evap./deposition/sublim.
  - Precipitation
  - Stability (Vertical/Slantwise Ascent)
  - Convection
  - Entrainment
- Radiative transfer
  - UV/visible/near-IR/thermal-IR
  - Scattering/absorption
  - Snow, ice, water albedos

- Meteorological processes
  - Velocity
  - Geopotential
  - Pressure
  - Water vapor
  - Temperature
  - Density
  - Turbulence
- Surface processes
  - Temperatures and water content of
    - Soil        Water        Snow
    - Sea ice    Vegetation    Roads
    - Roofs
  - Surface energy/moisture fluxes
  - Ocean-atmosphere exchange
  - Ocean dynamics, chemistry

**Five steps in the generation of a numerical model product**

**Observations**
- All models require obs from an area larger than their forecast domain
- Forecasts longer than 2-3 days require global data sets
- Global Telecommunications System (GTS) gathers and disseminates conventional data to nearly all countries

**Analysis**
- Objective analysis – obs checked for errors and interpolated to grid on which model atmosphere is represented

**Initialization**
- Adjusts the analyzed data so that the model and data are dynamically consistent
- Ensures no "noise" is generated when forecast begins

**Forecast**
- System of forecast eqns marched forward in time until desired forecast length is reached

**Output**
- Forecast maps produced and sent to users, including computations of many quantities not directly forecast by the model
- Forecasts verified to document model errors and biases in order to formulate improvements in the future.

[Figure]

**Feedback between SST and rain rate**

Cool SST

High SST

Promote
convection

Heat SST

Low SST

suppress
convection

Coupled data assimilation uses constraints to inhibit rainfall
over high SST regions, so salinity fields follow observations.

**Contribution to ocean data assimilation:**
**atmos / land hydrology → salinity budget**

[Figure]

HYCOM water flux into ocean (mm/day) and current (vector)

[Figure]

**Public DWA hydrology data**

[Figure]

**Meteosat-blended gauge data**

- Atmospheric data assimilation generates rainfall

- Over-land run-off feeds into river catchments, combined with satellite soil moisture

- Coastal river discharge is diffused and advected by winds, waves, turbulence and currents

- Salinity fields incorporate satellite and in-situ measurements, thus effects of upwelling

[Figure]

**Soil Moisture and Ocean Salinity Satellite (SMOS)**

**Examples of Amazon plume and Agulhas Current**

[Figure]

Sea Surface Salinity (psu) from OCEAN5 analysis on 20180913

PR

Hurricane path

Amazon discharge

[Figure]

Agulhas off PE

U current

SODA3    GODAS    OSCAR    ORA5

0.200

Shelf-edge
virtual buoy

**time-series since 2008: agreement for SODA3 & ORA5,
but GODAS & OSCAR show weakness & discrepancies**

**Vanishing sea ice**

**and shrinking beaches?**

[Figure]

[Figure]

**Comparison of past & future:**

**consistent values and trends?**

[Figure]

[Figure]

- PE shelf time series of ECMWF-ora4 hindcast and HAD-esm projected 1-50 m zonal current.

- Gauge measured and HAD-esm projected sea surface height at PE harbour.

For periods of overlap between observed and projected data, confidence can be determined according to means, variance, annual cycle, trends, and other metrics

**Local validation studies**

[Figure]

[Figure]

- Comparison of daily HYCOM model at nearest grid-point and: (left) sea surface height from tide gauge in western False Bay and (right) sea surface temperature from NOAA satellite; 2008-2015

**HYCOM ability to detect temp gradients within False Bay**

[Figure]

Dec12-Feb13 averages

Daily data 2008-2016

ASCAT ability to detect climate shift in False Bay

**Intercomparison of products:**
**3-day SST ~1 Jan 2013**

[Figure]

**.04 GHR OI SST**    **.01 GHR MUR SST**    **.1 HYCOM rean**    **.08 HYCOM oper**    **.1 GODAS rean**

Hi-res ocean reanalyses should converge over time, as optimal solutions are achieved for radiometer engineering, atmospheric correction, bias removal and coupled DA.

**Low-res pattern**

[Figure]

**Remote sensing of productivity**

[Figure]

What is the appropriate visible band product for marine productivity?

green-band chlorophyll?

Or

red-band fluorescence?

Influence of salinity, turbidity?

Why choose?

Use the reanalysis concept: all obs have value, so blend.

Retain multi-day time-scale for composite de-clouding.

**Local validation studies**

[Figure]

CRO site Apr-Dec 2012

obs = 0.67 mod + 5.47
$r^2 = 0.38$

obs temp 10 (C)
model temp 10 (C)

ADF site Mar 14 - Dec 17

obs = 0.48 mod + 8.5
daily $r^2 = 0.39$

obs temp 10 (C)
model temp 10 (C)

warm bias

- Inter-comparison of HYCOM model 10 m temperature and UTR data at Algoa buoys (y-axis).

[Figure]

Near-surface temp obs density

**Local validation studies**

[Figure]

Durban offshore

sat = 0.96 mod + 1.1
$r^2 = 0.91$

NOAA SST

Hycom T

[Figure]

Near-surface temp obs density

- Comparison at shelf-edge of Hycom and NOAA sea surface temp 2009-2015 off Durban.

**Local validation studies**

[Figure]

- Comparison of 3-hourly waverider buoy and W3 model significant wave height (2011-2013) near East London.

**Local intercomparisons**

Over the shelf, west of PE

Coastal gradient in SADCO ships data

[Figure]

SST   Zonal wind

cold bias

Seasonal cycle in satellite, reanalysis and coupled model

**Changes in variance over time**

[Figure]

Daily SST over the shelf west of PE:
Temporal variability dominated by seasonal cycle pre-2006.
Event scale fluctuations much greater thereafter.
Increased 'noise' from inshore upwelling, finally resolved.

**Globally available resources:**
**most starting in the satellite era**

| LABEL | DEFINITION | RESOLUTION | SOURCE |
|-------|-----------|-----------|--------|
| CCMP | Cross-calibrated multi-platform marine wind reanalysis | 25 km | Univ Hawaii APDRC |
| CFSr2 | Coupled Forecast System reanalysis v2 (ocean) | 30-50 km | Univ Hawaii APDRC |
| ECMWF5 | European Centre Medium-range Weather Forecasts | 30 km | Climate Explorer |
| GODAS | Global ocean data assimilation system (NOAA) | 50 km | IRI Clim. Library |
| HYCOM | Hybrid Coordinate Ocean Model | 10 km | Univ Hawaii APDRC |
| IPCC | Coupled model projections (HAD3esm, etc) | 100+ km | Climate Explorer |
| MERRA2 | Modern Era Reanalysis for Research and Applications | 50 km | NASA-giovanni |
| MODIS | Moderate-imaging Infrared Spectrometer | 1-4 km | IRI Clim. Library |
| NASA | National Aeronautics and Space Administration | 25-100 km (satellite) | NASA-giovanni |
| NCEPr2 | National Centers for Environmental Prediction | 180 km (reanalysis) | IRI Clim. Library |
| NOAA | National Oceanic and Atmospheric Administration | 50 km | IRI Clim. Library |
| ORA5 | Ocean Reanalysis v5 from ECMWF | 25 km | Univ Hawaii APDRC |
| SODA | Simple Ocean Data Assimilation (UMD, Carton) | 50 km | IRI Clim. Library |
| W3 | Wavewatch v3 ocean swell reanalysis | 50 km | Univ Hawaii PacIOOS |
| WHOI | Woods Hole Ocean Inst (surface fluxes) | 50 km | Univ Hawaii APDRC |

See also: https://reanalyses.org/ocean/overview-current-reanalyses

**Value of higher resolution and operational reporting**

- Products of monthly time-scale and spatial resolution > 0.5° can not resolve the shelf environment and its fluctuations
- HYCOM 0.1° resolves the coastal gradient, and shelf-edge eddies and rings at daily time scale
- CFSr2 and MERRA2 hourly reanalysis resolve diurnal variability at 0.5° resolution, ECMWF5 available at 0.3° resolution
- Confidence in these products is diminished by the scarcity of in-situ marine reports over the South African shelf
- Global ocean data assimilation will proceed with or without us
- More emphasis is needed on real-time measurement and reporting
- So satellite and model products are calibrated toward reality,
- And able to be trusted for use, not only in research, but in strategic decision-making

**What is the solution?**
**exchange of emails**

- **From: Mark R Jury  1 Feb 2019 To: T.Morris  <weathersa.co.za> [SAWS Marine Coordinator]**
- **QUESTION - I ask how SAWS interacts with SA marine scientists to pass on real-time ocean data collected in our EEZ, for operational and coupled model [assimilation and prediction]?**
- **ANSWER - …the team are planning to address these [non-reporting] issues, [possibly with parallel monitoring systems that duplicate those 'missing', such as harbour buoys and tide gauges].**
- **REPLY - …the IOC website shows that all SA marine platforms are off-line. Could you make it part of your group's responsibility to get the data back online?**

- **3 Feb 2019 To: C.Rautenbach <weathersa.co.za> [SAWS Marine Dept]**
- **QUESTION - …path to operational oceanography. I was wondering what existing [marine] data could be [fed] to ocean models used in coupled forecasts? SAEON has quite a few buoys, and could report through SAWS GTS? Same for any data [reaching] SADCO…**
- **ANSWER - …we are having negotiations to add a variety of marine data to our regular GTS reports**

- **21 Feb 2019 To: K.Wilmer-Becker <metoffice.gov.uk> [GODAE Programme Coordinator]**
- **QUESTION – I want to know how much of South Africa's marine observations are reaching the global ocean DA system on an operational basis?**
- **26 Feb 2019  ANSWER - CMEMS Service Desk <mercator-ocean.eu>**
  **[According to the] Copernicus insitu expert team …we don't have [any] platforms identified as coming from South Africa [over the most recent DA cycle]: 7 ships, 2 drifting buoys, 4 argo floats, 12 tagged fish. All of them are moving platforms [of external origin]…**
- **REPLY - …in case of 'privatized' data that requires confidentiality… is it possible to 'flag' reports, so to assimilate but keep actual data 'hidden'?**
- **ANSWER - Yes, that is an option many services are using: [ECMWF, UKMET, METEO-FRANCE, etc].**

**How we solve the problem in Puerto Rico**

- NOAA contracts the university to provide operational data monitoring and real-time reporting (CARICOOS)
- Graduate students engaged to do much of the work under supervision of professors
- Contract is on-going and stipulates 99% data capture, has budget for maintenance, replacement equipment, bursarys / internships.
- All data are required to be publicly available within 1 hour of collection, mirrored on government websites, with data QC and assimilation handled by quasi-government ocean, land & weather services and NOAA subsidiaries (US-Navy, USGS, etc).
- The university does not conduct in-house data assimilation, that is the work of major centers, given the need for blending with vast amounts of NASA satellite measurements. We maintain the observing & reporting system, and evaluate / validate the DA model outputs.

**Puerto Rico's operational system**

Marine & coastal data collection & reporting by University of Puerto Rico Mayaguez

Caricoos marine monitoring system

[Figure]

IOC mirror site

Actual and virtual buoys

[Figure]

**Puerto Rico's operational system**

PR economy similar to South Africa, except many jobs are automated, all data are publicly available.

[Figure]

NDBC moored buoy + wx station, with hourly real-time data since 2000+

University researchers assist gov. operations via graduate intern field work, identification of 'bad' obs & systematic DA model errors, applied research theses, publications, outreach.

[Figure]

NDBC buoys around Puerto Rico

[Figure]

HF radar currents

[Figure]

Waves

[Figure]

Winds

nested model outputs – daily update downscaled from operational products

Tide gauge maintained since 1962

**Puerto Rico's operational system**

[Figure]

**Both ocean and weather services feed to GTS, to ensure real-time reporting to global centers for data assimilation, within atmospheric cut-off window (3 hr)**

**Real-time reporting weather stations, via Wundermap, with 10 minute update, most derive from quasi-government services**

[Figure]

[Figure]

USGS 50147800 RIO CULEBRINAS AT HWY 404 NR MOCA, PR

△ Median daily statistic (50 years) — Discharge

**Daily Streamflow Conditions**

[Figure]

**Real-time reporting streamflow gauges, with 10 minute update: the flash-flood warning system linked to wx radar**

**Cabo Rojo Puerto Rico  May 2019**

| Day of Month | Day of Year | Total Solar Rad. ly. | Wind Ave. V. mph | Dir. Deg | Max. mph | Air Temperature Mean Deg. Fahrenheit | Max | Min | Humidity Mean Percent | Max | Min | Dew Point Deg. Fahrenheit | Wet Bulb | Total Precip. inches | Total Penman ET. inches |
|---|---|---|---|---|---|---|---|---|---|---|---|---|---|---|---|
| 1 | 121 | 594 | 14.5 | 103 | 37.0 | 82 | 89 | 76 | 67 | 79 | 50 | 69 | 72 | 0.00 | 0.29 |
| 2 | 122 | 460 | 13.7 | 103 | 32.0 | 81 | 87 | 77 | 68 | 81 | 56 | 69 | 72 | 0.00 | 0.23 |
| 3 | 123 | 514 | 13.0 | 96 | 30.0 | 80 | 86 | 73 | 64 | 78 | 49 | 67 | 70 | 0.00 | 0.25 |
| 4 | 124 | 513 | 12.8 | 106 | 27.0 | 80 | 86 | 75 | 64 | 74 | 50 | 67 | 70 | 0.00 | 0.26 |
| 5 | 125 | 573 | 16.3 | 105 | 35.0 | 81 | 87 | 76 | 67 | 75 | 54 | 69 | 72 | 0.00 | 0.28 |
| 6 | 126 | 573 | 15.2 | 108 | 30.0 | 81 | 86 | 76 | 70 | 76 | 57 | 70 | 73 | 0.00 | 0.27 |
| 7 | 127 | 590 | 15.0 | 106 | 33.0 | 81 | 85 | 76 | 68 | 76 | 57 | 69 | 72 | 0.03 | 0.27 |

**Value of coupled forecasts**

- We used to think that coupled modelling was needed only for long-range predictions (> 2 month lead time) driven by alternating ENSO phase and accumulating greenhouse gases.

- With the advent of hourly-fluctuating, eddy-resolving ocean and land products, it is evident that short-range predictions (> 2 day lead time) out-perform uncoupled forecasts.

- Coupled models better simulate the diurnal cycle of rainfall and wind speed*, changes in tropical cyclone intensity, etc.

- There is a single assimilation system for environmental data; and converging model technology for land, ocean, atmosphere (both physical and chemical).

- Long-range predictions for South Africa summer climate show increasing skill, via ENSO influence on slow undulations of the ocean thermocline that modulate atmospheric convection and circulation.

- But model-simulated fields, mainly derived from satellite estimation, need local calibration (error-constraints) for operational use.

[Figure]

ECMWF5 max temp Dec 2018

hourly winds 31 Dec 2018 GAMTOOS MOUTH

CFSr2 (red) vs station

---

## Author Comment (AC5) · 13 Oct 2020

**Marine climate change over the eastern Agulhas Bank of South Africa**

Mark R Jury

Geography Dept, University of Zululand, KwaDlangezwa 3886, South Africa and Physics Dept., University of Puerto Rico Mayaguez, USA, 00681

re-submitted to EGU ocean Sci Oct4 June 2020

**Abstract**

The rate of change in the marine environment over the eastern Agulhas Bank, along the south coast of South Africa (32-37S, 20-30E) is studied using reanalysis observations 1900-2015 and coupled ensemble model projections 1980-2100. Outcomes are influenced by resolution and time-span: ~1 degree datasets covering the whole period capture large-scale changes, while ~0.5 degree datasets in the satellite era better distinguish the cross-shelf gradients. Although sea surface temperatures offshore are warming rapidly (.05°C/yr since 1980), a trend toward easterly winds and a locally stronger Agulhas Current have intensified near-shore upwelling (-.03°C/yr). The sub-tropical ridge is gradually moving during summer is drawn poleward, leading to a by global warming and high phase southern oscillation index. Cooler inshore sea temperatures suppress latent heat flux and contribute to coastal desiccation (-.005 mm day$^{-1}$/yr) and vegetation warming (.1°C/yr) since 1980. Coupled ensemble projections from the Hadley and European models indicate that the shift toward drier climateweather and easterly winds may be sustained through the 21$^{st}$ century.

Key words: South African, coastal climate change mark.jury@upr.edu

**Introduction**

The marine climate of the eastern Agulhas Bank along the south coast of South Africa is shaped by the continental plateau and sub-tropical latitude. Rainfall tends to be limited and shelf waters are characterized by sharp gradients between inshore upwelling and an offshore current that advects warm water polewards at ~1 m/s (Lutjeharms et al. 2000). Downstream widening of the shelf and cyclonic shear causes uplift at the shelf edge (Schumann et al. 1982, Lutjeharms 2006, Goschen et al. 2015, Malan et al. 2018). Westerly and easterly wind regimes during winter to summer respectively induce alternating spells of downwelling and upwelling (Schumann and Martin 1991, Schumann 1999). Numerous small rivers discharge into the shelf zone (Schumann and Pearce 1997, Scharler and Baird 2005, vanBladeren et al. 2007). The inshore environment and large embayments (Fig 1a) are characterized by weak circulations and seasonal warming, and become stratified and productive during austral summer (Roberts 2010, Pattrick et al. 2013). The Agulhas Current meanders a few times per year (Goschen and Schumann 1990, Rouault and Penven 2011), while the mid-latitude jet stream meanders a few times per month advecting coastal lows and continental shelf waves along the shelf (Jury et al. 1990, Schumann and Brink 1990). Amidst these rapid changes are rising sea levels (Mather et al. 2009; Jury 2018) and longer summers.

The eastern Agulhas Bank shows trends toward offshore warming and inshore cooling due to wind- and current-induced upwelling, and retreat of the circumpolar westerlies (Rouault et al. 2009, Durgadoo et al. 2013, Hutchinson et al. 2018). Trends in air temperatures are near the global average Past research has found trends of .02°C/yr (Kruger and Shongwe 2004, Morishima and Akasaka 2010, Jury 2013), buthowever trends in other variables show multi-year fluctuations (Philippon et al. 2012) from tend to be over-shadowed by regional atmosphere coupling with sea surface temperatures (SST) and the Pacific El Niño Southern Oscillation (ENSO). Climate changeshort-term analyses of events and the sparsity of data before 1950 (Tadross et al. (2005),

MacKellar et al. (2014) and Kruger and Nxumalo (2017) offer guidance on resource management, which this research seeks to extend.

The main objective of this study is to establish the rate and pattern of observed and projected marine climate (land, air, sea) trends along the south coast of South Africa from 1900 to

2100. Scientific questions include: 1) How has the wind field responded to a poleward shift of the subtropical ridge? 2) What are the con- sequences of intensified  coastal  upwelling

? 3) How does record length and dataset resolution affect the result? and 4) What is the impact of climate variability  on trend attribution? While the spatial focus is on the south coast of South Africa using monthly datasets finer than 0.5° during the satellite era, context is provided at the large-scale using coarser model products over the 20[th]

and 21[st] centuries.

**Data and Method**

Modern data assimilation systems blend in-situ and ancillary measurements by iterating be- tween climatology, persistence and theory, interpolating across gaps in time and space, and limiting the influence of outliers. By reducing uncertainties, scientists now have a reliable means to evaluate trends in marine climate. The monthly reanalysis products employed here include: ECMWFv5

coupled (Dee et al. 2011), ECMWF-20c atmosphere (Poli et al. 2016), ECMWF-ora4

ocean (Balmaseda et al. 2013), NASA MERRA-2 coupled (Gelaro et al. 2017), NCEP

CFSr-2 coupled (Saha et al. 2010), SODA-3 ocean (Carton et al. 2018), NOAA sea surface temper- ature (SST; Reynolds et al. 2007), NOAA net outgoing longwave radiation (OLR; Lee et al. 2007),

NESDIS vegetation temperature (Tucker et al., 2005). and CHIRP rainfall (Funk et al. 2014).

Table 1 lists the acronyms, data source, horizontal resolution and time-span. Ocean-atmosphere fields with horizontal resolution finer than 0.5° are capable of representing cross-shelf gradi- ents, and these are available in the satellite era 1980-2016. SODA-3 provides sub-surface ocean data on temperature, salinity, currents and vertical motion; driven by MERRA-2 winds, multi- satellite altimeter and thermal measurements, blended with in-situ observations over the shelf.

Coupled and-atmosphere-ocean evolution is described by coupled reanalysis  products underpinned by data assimilation (Hamrud et al 2015).

In addition to the monthly datasets, daily ECMWF-5 sea level air pressure (SLP) fields were analyzed using  empirical orthogonal functions (EOF). The leading mode was deter- mined and its spatial loading pattern and time score were analyzed for evolution at lags from -2 to

+2 days, and for trends and spectral cycling in the period 1900-2015. Ship data, from the repos- itory for marine data collected in South African waters: SADCO, were analyzed in 0.1° bins for

SST and wind speed, averaged 24.5-26.5°E 1950-2015 (cf. Fig A1) and compared with 0.3° reanal- ysis products. Monthly river discharge records were obtained for the Gamtoos and Sundays Rivers from the SA Department of Water Affairs hydrology service: SADW, and combined to understand the coastal hydrology.

Bias was examined via inter-comparisons between SADCO SST and wind speed, and the satellite-era reanalyses (CFSr-2, ECMWF-5, MERRA-2). These show coherent cross-shelf gradi- ents (cf. Fig A1) indicating they capture the inshore upwelling. The reanalyses diverge at the coast, depending on resolution and land-sea ratio.

[revised manuscript text omitted]

**Shelf analysis and gradients**

Hovmoller plots were constructed across the southern shelf (Fig 4a-d) for 18-month filtered SST  zonal winds / currents / vertical motion, and rainfall. There is a multi-year alternation of warm and cool .  spells

, modulated by local winds and the Pacific El Nino / La Nina (Jury 2015, 2019) and the Southern Annular Mode (Malan et al. 2019). T there is a background trend of inshore cooling and offshore warm- ing that intensifies the coastal gradient (Fig 4a). The SST pattern is supported by Ekman transport from inshore easterlies and offshore westerlies - that pulse in 1992 and 2013 (Fig 4b). Rainfall (Fig

4c) displays a sharp boundary at 34.5°S between dry inshore / wet offshore climates. Coastal upwelling and atmospheric subsidence suppress moist convection, whereas the Agulhas Current enhances marine rainfall ~ 3-fold. The sharp change in CHIRP rainfall regime on 34.5°S coincides with accelerated longshore winds. The hovmoller plot of SODA-3 near-surface zonal currents (Fig

4d) reveals pulsed intensification and coastward shift, contributing to near-shore uplift > 4 m/day (34.1-34.4°S). C current- and wind-induced upwelling appear additive much of the time. How- ever in 2013 currents prevailed over winds, suggesting occassional decoupling.

Index-area time series of reanalysis and projected near-surface zonal currents (Fig 4c) show a trend of local -acceleration. Past and future linear regression slopes are -.0076 m s$^{-1}$

/yr, with trend correlations rising from -.81 to -.90. Future (2$^{nd}$ order) trends overlie those from past reanalysis and year-to-year fluctuations are consistent despite technology artifacts of satellite altim- etry and ensemble averaging. Appendix A2 compares the index-area annual cycle of model vs ob- servation. This index-area covers much of the Agulhas Current in longitudes where the shelf is con- vex (cf. Figs 1a, 2e).

The trend of NOAA SST analyzed in coastal and shelf-edge latitudes show contrasting val- ues but little change over the annual cycle in Fig 5a. Shelf-edge waters are warming steadily (r=

+.5) while coastal waters are cooling (r= -.5) moreso from February to May (slope -

.04°C/yr). Together these indicate a tightening gradient ($\partial T/\partial y$) and a steepening sea slope. The annual cycle of index-area zonal wind trends (Fig 5b), averaged over three reanalyses, reveals that easterly winds are intensifying during summer (Nov-Feb), when subtropical ridging is most likely.

Regression of SST and winds onto the southern oscillation index (Fig 5c,d) reveals trend patterns similar to climate change: inshore cooling (mainly summer) and offshore warming (all- year). Winds with respect to high-phase SOI are from northeasterly and considerably stronger in summer, hence wind-driven coastal upwelling is favoured during La Nina. The southern oscillation index has shown an upward trend during the satellite era, and its regression onto regional sea level air pressure patterns (cf. Fig A3) matches the earlier mode-1 pattern of mid-latitude high / sub- tropical low (cf. Fig 3c). Hence long-term and multi-decadal trends tend to conspire

.

**Hydrology trends**

The  increasing near-shore salinity (cf. Fig 2b) could be  related to drying trends in the adjacent terrestrial climate (cf. Fig 3b).

In Fig 5e the regional hydrology is studied using the combined Gamtoos and Sundays River dis- charge record. Although flood / drought events and 2-5 yr cycles are evident, there is little trend. The study area lies between a zone of reduced cloudiness (Benguela – Namib) to the northwest and increased cloudiness to the southeast, as seen in the trend map for satellite net OLR

(Fig 5f). The rising  salinity off the south coast (cf. Fig 2b) may be a to advection from the Mozambique Channel, where evaporation exceeds precipitation (cf.

Fig 1e). Vertical motions over the shelf could also play a role (cf. Fig 2f), whereby cyclonic shear lifts salty water.

**Model projections under greenhouse warming**

Spatial maps of ECWMF-esm rcp8.5 trends for zonal wind and rainfall 1980-2100 show a key feature southeast of the study area (Fig 6a,b). Easterly winds are projected to increase and rainfall is expected to decrease. The warm moist air carried westward beneath a stable inversion layer generates less evaporation, so rain-bearing storms are projected to diminish in strength and be deflected poleward by the sub-tropical anticyclone.

Time series of index-area values comparing ECMWF-20c reanalysis with ECMWF-esm and Hadley-esm projections are given in Fig 6c-f. Coupled ensemble values overlie the observa- tion-based product indicating little bias but lower variance. Zonal winds that oscillate in a station- ary manner through the 20[th] century tend toward easterly (-U) in conjunction with declining pre- cipitation. Air temperatures show a gradual rise during the 20[th] century in both reanalysis and overlapping simulation. Thereafter, the warming trend steepens due to the greenhouse scenario.

There appears to be little moderating influence of cooler nearshore SST, which coarse resolution products under-represent (cf. Fig 1c). The SOI time series is relatively stationary, but larger am- plitude swings are noted in the early 20[th] and late 21[st] century. High phase (Pacific La Nina)

events seem steady but El Nino events appear to deepen after 2040. In summary, past zonal winds of 1 m/s (after cancellation of east-west components) are projected to reach -1 m/s by 2050. Past rainfall of 1.5 mm/day declines below 1 mm/day, and air temperatures of 17°C rise above 20°C

by 2050. The regression $r^2$ fit of trends are in the range from 72-97% and suggest sustained changes for temperature, however wind and rain tend to oscillate in the ECMWF-esm projection until the rcp8.5 scenario prevails.

In addition to ENSO influence, the Southern Annular Mode (SAM) plays a role in the lati- tude and intensity of basin-scale anticyclonic gyres that support the Agulhas Current (Yang et al.

2016; Elipot and Beal 2018). The long-term trend in the SAM is a contraction of circumpolar westerlies that enables poleward expansion of the tropical Hadley circulation and belt of easterly winds rounding the tip of Africa seen here (cf. Fig 3a, 6a). Yet SAM trends are flattening with recovery of the Antarctic ozone hole (Arblaster et al. 2011), and may exert less effect in future.

**Discussion and summary**

This study addressed a range of questions around spatial patterns in trends and uncovered evidence of a pulsed poleward shift of the subtropical ridge (cf. Fig 3c,d). Analysis of land- atmosphere-ocean conditions revealed intensified coastal upwelling from increased easterly winds.

A steeper ∂T/∂y produces aa locally faster shelf-edge current, with consequences for and currentinduced upwelling (Schumann & Beekman 1984, Swart & Largier 1987) and  coastal dessication. Employing coupled reanalysis and model projections to distinguish coast and offshore features, a unifying pattern emerged: summer-time wind-driven upwelling enhances geostrophic gradients and the Agulhas Current. Although ocean reanalysis outcomes are moving toward  consensus based on a shared data assimilation system,  interpretations need not favour one process over another: wind vs current, fluxes vs advection, multi-decadal vs trend, local vs remote. Multi-variate forcing is  expected .

To place these results in context, trends in global  SST were analyzed over the satellite era (not shown). Coastal upwelling zones show cooling < -.03°C/yr: broadly Peru 5-25S and California 30-40°N, and narrowly off Somalia 10-15N, Namibia 35-20°S and Western Sahara 15-30°N. Even shelf waters of the USA Carolinas 30-40°N are cooling and, like the south coast of South Africa, there is a warm current offshore. Steeper gradients could produce faster shelf-edge flow, but the Gulf Stream is decelerating (Jury 2020) unlike the Agulhas Current. Figures 2c and 4d gave evidence of locally  increasing westward currents off Cape St Francis  ocean reanalysis and coupled model projections. Perhaps wind-driven eddies are broadening the Agulhas Current over  multi-year periods, in addition to background  trends (Elipot and Beal 2018). International monitoring efforts such as the ASCA line (Morris et al. 2017) could resolve ambiguities arising from the extrapolation of short-term records. Our analysis does not claim the whole Agulhas Current is strengthening, only along the shelf-edge of the eastern Agulhas Bank.

Another way of placing these results in perspective is to compare trends in coastal SST with variance from the annual cycle (i), inter-annual variability (ii), and intra-seasonal fluctuations (iii). The index-area standard deviations are: 2.5°C (i), 0.7°C (ii), and 0.9°C (iii) respectively, compared with a 35-yr decline in coastal SST of -2.4°C. Applying linear regression to coastal SST data with and without the annual cycle achieves r = -.29 vs -.76. Either way the trend is significant, not only statistically but in terms of environmental impact.

In this study, modern reanalysis datasets have been used for mapping the marine climate trends over the southern shelf of South Africa. Cross-shelf gradients in sea temperatures, latent heat flux, currents and upwelling are apparent in the satellite era. SST in the offshore zone is warm- ing (.05°C/yr) since 1980 and there is a trend toward easterly winds, mainly in summer (U = -.015

m s$^{-1}$/yr). The shelf-edge Agulhas Current is accelerating (U = -.006 m s$^{-1}$ /yr) in longitudes 21-28E

(cf. Fig 2c) partly due to large scale winds over the southwest Indian Ocean (Backeberg et al. 2012)

that align with the local forcing seen here. The faster current and 'following' wind induces coastal uplift (Leber et al. 2017) and cooling (-.03°C/yr). As the sub-tropical ridge is drawn poleward, the cross-shore gradient steepens (cf. Fig A1). Cooler near-shore sea temperatures correspond with atmospheric subsidence, drying trends (-.005 mm day$^{-1}$/yr) and vegetation warming (.1°C/yr). Similar trends in local air-sea interactions are attributed to more frequent wind-driven coastal upwelling and easterly winds (cf. Fig 3a) similar to  Malan et al. (2019). Coupled ensem- ble projections from the Hadley and European models indicate that the shift toward drier weather, easterly winds, coastal upwelling and a locally faster Agulhas Current may be sustained through the

21$^{st}$ century (cf. Fig 6c), as a local response to the poleward shift of the sub-tropical ridge. Some of the environmental changes could  create opportunities for resource adaptation (Jury 2019) and spark interest in aquaculture and ecotourism.

While the shelf may benefit, terrestrial water resources could be headed towards greater stress. Although the hydrology is transitionally located between a drying west and moistening east, the Sundays River sees inter-basin transfers while the Gamtoos River depends on agricultural 're- cycling'. In both cases reduced runoff linked to rainfall could inhibit freshwater fluxes to the coastal ocean (cf. Fig 6b).

Parallel work on this geographic niche (Jury 2019, Jury and Goschen 2020) is on-going and further studies will: i) compare observation and reanalysis trends, ii) consider how changing satel- lite technology represents shelf dynamics, iii) quantify wind- vs current-driven upwelling, and iv)

analyze coupled models capable of detecting sharp coastal gradients.

**Acknowledgements**

SAPSE funding support from South Africa is acknowledged. Reanalysis and projection Most data derive from websites of the IRI Climate Library, KNMI Climate Explorer and Univ Hawaii AP-

DRC.

[revised manuscript text omitted]

Figure 4  Hovmoller plots of 18-month filtered variables avg. 24.5-26.5E: (a) NOAA SST (C), (b) zonal
wind stress (N/m²), (c) CHIRP rainfall (mm/month), (d) SODA-3 1-50 m zonal current (shaded m/s) and
1-200 m upward motion (contour m/day). (e) Index-area time series of observed and projected 1-50 m
zonal current. 'Coast' and 'Shelf' climates and average wind speeds are labelled in (c).

[Figure]

Figure 5 Analysis of monthly index-area trends for (a) coastal and shelf-edge SST, and (b) zonal wind and its significance (1-p value), with 35 degrees of freedom. Regression of (c) annual and (d) summer NOAA

SST (shading °C) and SODA-3 surface wind (vector, scale inset m/s) with the SOI index 1981-2016 (units are per SOI fraction). (e) Observed discharge of the combined Gamtoos and Sundays Rivers. (f) Trend of

NOAA net outgoing longwave radiation as a proxy for cloudiness (W m$^{-2}$/yr 1979-2017) with dot showing river gauges.

[Figure]

rea

Figure 6 EC-esm projected trend maps 1980-2100: (a) zonal wind (m s$^{-1}$/yr), (b) precipitation (mm day$^{-1}$/yr).
Temporal record of index area ECMWF-20C reanalysis 1900-2010 and EC-esm projected 1980-2100: (c)
zonal wind, (d) precipitation, and (e) air temperature. (f) Observed and model projected Pacific southern
oscillation index (east-west SLP EOF mode). Best-fit trends are given; time series are composed of annual
averages.

**Appendix**

[Figure]

Fig A1  SADCO ship data, averageds per in each 0.1 latitude bin over 24.5-26.5E longitude 1950-2015; left axis and dashed line refer to standard deviation; (line) and comparativeison with satellite era 0.3 reanaly-sisbinned CFSr2 (left) and ECMWF (dots).

[Figure]

Fig A2  Annual cycles averaged over the index area; comparing model SST, surface zonal wind (middle) and near-surface current with reference product. The model has an amplified annual cycle that is cooler and more westerly in winter. Currents show summer / winter regimes with model slightly weaker and delayed.

[Figure]

Fig A3  Graph of 18-month filtered southern oscillation index and its trend in the satellite era, and regression of Dec-Feb SOI onto regional sea level air pressure (hPa), with boxed index area.

---

## Referee Comment (RC4) · Ricardo Matano (Referee) · 26 Oct 2020

It was a pleasure to read this interesting manuscript about the impact of climate related changes on the Agulhas Bank. Using ECMWF re-analysis products, coupled model simulations and satellite data Dr. Jury analyzes the impact of climatic trends on oceanic variables and speculates about the potential impacts of these trends on the regional conditions. This information will not only be useful to earth scientists and students but also to policymakers.

The main criticism that can be made to this article is that although it concerns the impact of climate change on the ocean state, it does not seriously consider how changes in the large-scale oceanic circulation affects the shelf area. Instead, it focus all the

attention on atmospheric variables. I think that the article would benefit in considering how climate related changes of the Agulhas Current might affect (through shelf/deep-ocean interactions) the shelf region. It seems reasonable to surmise that the reported (Backberg et al., Elipot and Beal, etc) intensification of eddy variability in the Agulhas C. should lead to similar changes of shelf/deep-ocean exchanges. Those exchanges are not small; in our own simulations we have observed exchanges of the order of 1-10 Sverdrups (although the along shelf transports are one order of magnitude smaller!). To put these values in context it is useful to compare them with the fdischarges of local rivers, which are of the order of 1-10 m3/s. That is, shelf/deep-ocean exchanges are between one million to one billion times larger than river discharges, yet these exchanges are largely ignored. My specific recommendation to Dr. Jury, therefore, is to include an analysis and discussion of these matters. I think that this inclusion will strengthen the analysis and the appeal of this article.

I found intriguing the lack of inclusion of satellite sea surface height (SSH) data in the analysis. From a dynamic point of view SSH is far more important that SST data, since it represents a deeper portion of the water column. Use of SSH data, moreover, would allow Dr. Jury to link changes in the atmospheric and oceanic circulation. I am a bit skeptical, for example, of the conclusion that a tightening of the SST gradient should necessarily lead to a steepening of the sea slope (1st paragraph of page 9).

In summary, I think that this is a very nice article that can be substantially strengthened with a more in depth analysis of the trends in the large-scale circulation and their impact on the shelf region.

---

## Author Comment (AC6) · 26 Oct 2020

The author agrees with this reviewer, and added a paragraph at the end of the hydrology section on cross-shelf exchanges. An analysis was added (Appendix Fig 4a,b) that considered the sea surface height in a cross section south of Port Elizabeth, and in time series that compared past and future sea level rise. The analysis confirms (using GODAS 0.1deg hindcast) that the SSH slope is steepening offshore due to the inshore cooling and offshore warming. Additional information was added in the Data section, in the Data Table 1, in the Discussion pg 13, and a data reference was added. The paper was updated with these changes and will be uploaded as v2 (with track changes, revised final figure).

---

## Editor Comment (EC1) · Piers Chapman (Editor) · 2 Nov 2020

The paper is now generally looking good, although on p7, section 3.4, the final paragraph refers to Fig 4c on zonal currents. As Fig 4c shows the CHIRP rainfall data, I think you mean either Fig 4d or 4e. (This error is carried forward from the previous version of the manuscript).

My only other comment refers to another point made by reviewer #4. You have added a statement on p.8, section 3.5, about the reported intensification of eddy variability and the likely increased cross-shelf exchanges that result. But the question then is, so what? How will these affect the local hydrology and possible trends in your study area?

Yang et al 2016 is not in the reference list.

---

## Author Response (AR1)

Universidad de Puerto Rico
**RECINTO UNIVERSITARIO DE MAYAGUEZ**
Mayagüez, Puerto Rico 00681-9016

*Facultad de Artes y Ciencias*

**Departamento de Física**

[Figure]

University of Puerto Rico
**MAYAGUEZ CAMPUS**
Mayagüez, Puerto Rico 00681-9016

*College of Arts and Sciences*

**Department of Physics**

30 October 2020

Editor of Ocean Sciences

**Inclusion of reviewer feedback**

In earlier submissions both online and to the topic editor, I have provided all responses and track-change versions. The latest version is formatted in journal style with figures embedded.

Sincerely,

Professor Mark R. Jury
mark.jury@upr.edu